# Decoding Open-Ended Information Seeking Goals from Eye Movements in Reading

**Cfir Avraham Hadar,**[*] **Omer Shubi,**[*] **Yoav Meiri, Amit Heshes, Yevgeni Berzak**
Faculty of Data and Decision Sciences, Technion - Israel Institute of Technology, Haifa, Israel
`{kfir-hadar,shubi,meiri.yoav,amit.heshes}@campus.technion.ac.il`
`berzak@technion.ac.il`

## Abstract

When reading, we often have specific information that interests us in a text. For example, you might be reading this paper because you are curious about LLMs for eye movements in reading, the experimental design, or perhaps you wonder "This sounds like science fiction. Does it actually work?". More broadly, in daily life, people approach texts with any number of text-specific goals that guide their reading behavior. In this work, we ask whether open-ended reading goals can be automatically decoded solely from eye movements in reading. To address this question, we introduce goal decoding tasks and evaluation frameworks using large-scale eye tracking for reading data in English with hundreds of text-specific information seeking tasks and auxiliary annotations of task-critical information. We develop and compare several discriminative and generative multimodal text and eye movements LLMs for these tasks. Our experiments show considerable success on selecting the correct goal among several options, and even progress towards free-form textual reconstruction of the precise goal formulation. We further tie model performance to cognitively interpretable aspects of human gaze behavior. These results open the door for further scientific investigation of goal driven reading, as well as the development of educational and assistive technologies that will rely on real-time decoding of reader goals from their eye movements.[1]

## 1 Introduction

Eye movements in reading are a key methodology for studying the cognitive basis of reading and language processing. When we read, our eyes move over the text in a saccadic manner, with prolonged periods of time in which the gaze is stable, called *fixations*, and fast transitions between fixations called *saccades* (Schotter & Dillon, 2025). This trajectory contains rich behavioral traces of the ways in which readers interact with texts and process language in real time (Just & Carpenter, 1980; Rayner, 1998). Understanding the nature and strength of the relation between eye movements, the text, and online language processing has been a major research avenue in the psychology of reading in the past few decades, and in recent years has also been gaining increasing interest in NLP and machine learning (Barrett & Hollenstein, 2020; Reich et al., 2025).

In this work, we investigate a reading scenario that is prevalent in daily life but remains understudied – *seeking specific information in a text*. This scenario deviates from a common implicit assumption in psycholinguistics, according to which the comprehender's goal is constant across communicative contexts: a general understanding of the linguistic input. In practice, readers approach texts with a variety of goals and information seeking needs. Each such goal can have a profound impact on the cognitive processes of language comprehension and the corresponding reading behavior (Radach & Kennedy, 2004; Kaakinen & Hyönä, 2010; Hahn & Keller, 2023; Shubi & Berzak, 2023).

Our study focuses on the following question: can *arbitrary, text-specific information goals* be automatically decoded from eye movements in reading? Specifically, given a single participant reading a single passage with a question in mind that they would like to answer via the text, can this question

---

[*]Equal contribution.
[1]Code is available at https://github.com/lacclab/Open-Ended-Goal-Decoding.

be decoded from their eye movements over the passage? This task has not been addressed in prior research and has both scientific and practical importance. Scientifically, it allows obtaining insights into the strength and nature of the relation between task conditioning and reading behavior, and more broadly, the extent to which eye movements contain information on the cognitive state of the reader. Practically, it opens the door for applications in education, content personalization, and text accessibility, which will rely on detecting information seeking behavior, its specific purpose, and the extent to which this behavior is effective.

Our key contributions are the following:

**Tasks:** We present a new cognitive state decoding challenge from eye movements in reading: decoding arbitrary information seeking goals over the content of specific texts. We instantiate this challenge as a goal *selection* task given several possible goals, and as an even more challenging goal *reconstruction* task where the information-seeking goal has to be generated.

**Models:** For the goal selection task, we introduce two strong baseline models and adjust two state-of-the-art *discriminative* models of text and eye movements (Haller RNN and RoBERTEye). For the goal reconstruction task, we develop and finetune two *generative* LLMs, with textual representations of eye movements (DalEye-Llama, DalEye-GPT). We further use textual eye movement representations to address this task using a frontier LLM (Gemini) in zero-shot and few-shot regimes.

**Evaluations:** We perform systematic evaluations of goal selection accuracy at two levels of task difficulty and utilize several generation quality measures for the goal reconstruction task. For both tasks, we examine model generalization across new texts and new readers. Finally, we tie model performance to cognitively interpretable properties of human gaze behavior during goal decoding.

## 2 EYE TRACKING DATA

We use OneStop (Berzak et al., 2025a), a public eye tracking dataset collected with an EyeLink 1000 Plus eye tracker. It comprises 360 adult native English speakers reading Guardian articles from the OneStopQA multiple choice reading comprehension dataset (Vajjala & Lučić, 2018; Berzak et al., 2020). In each experimental trial, a participant reads a paragraph on a screen and, on the following screen, answers a multiple-choice reading comprehension question, without the ability to return to the paragraph. Each paragraph can appear in its original or human-simplified form. 180 participants read in an *information seeking* condition, where they receive the question (but not the answers) before reading the paragraph, and therefore have a specific *reading goal* for the upcoming paragraph. The remaining 180 participants do not receive the question ahead of time, and thus have to be ready to answer any subsequent question. We use the information seeking part of OneStop (Berzak et al., 2025b), which is uniquely suited for our study.

For each paragraph, a participant receives one of three possible questions over the paragraph. Each such question has a manually annotated *critical span* in the paragraph, which contains the information that is essential for answering the question. The questions tend to be more inferential than extractive: typically, the critical span does not contain the answer in verbatim form, but rather the essential information from which it can be inferred (Berzak et al. (2020), and see Appendix A for an empirical analysis which supports this property). Two of the questions share the same critical span, while the third question has a distinct critical span. This design supports a question selection task with two tiers of difficulty, distinguishing between questions with non-overlapping critical spans and the more challenging variant of distinguishing between questions over the same critical span. Figure 1 presents an example of a paragraph, its three questions, and their critical span annotations.

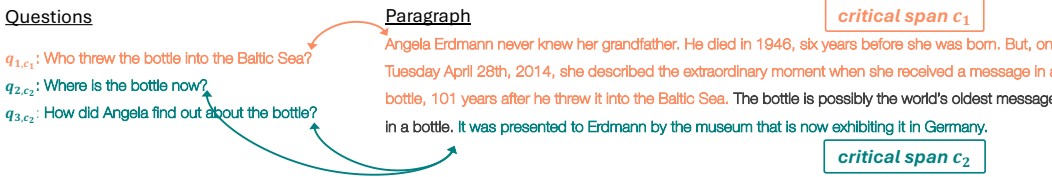

Figure 1: Example of a OneStop paragraph, its three questions, and each question's *critical span*: the paragraph segment essential for answering the question. See details on notation in Section 3.

OneStop Information Seeking has 486 questions over 162 paragraphs, with 20 participants answering each question. The mean question length is 10 words. The mean paragraph length is 109 words (5.8 sentences), of which the critical span is 32 words on average. Overall, there are 1,055,429 paragraph word tokens for which eye tracking data was collected.

## 3 PROBLEM FORMULATION

The general problem we address is using eye movements over a text (in our setting, a text is always a paragraph) to decode which question a participant seeks to answer using that text. Each text $T$ has three *text specific* questions, $Q_T = \{q_{1,c_1}, q_{2,c_2}, q_{3,c_2}\}$. The critical information for answering question $q_{1,c_1}$ is located in the text in span $c_1$, while for both questions $q_{2,c_2}$ and $q_{3,c_2}$ it is in span $c_2$. In each experiment trial, a participant $P$ is presented with one question, $q_P \in Q_T$, before reading the text. The recording of the participant's eye movements while reading the text is denoted by $E_{P,T}$. We propose the following two task variants.

**Goal Selection** In the goal selection task, a classifier $h$ has to *select* which question from $Q_T$ the participant received, from their eye movement recording over the text:

$$h(T, E_{P,T}, Q_T) \longrightarrow \hat{q}_P \in Q_T$$

**Goal Reconstruction** In the goal reconstruction task, a generative model $g$ has to *generate* the question that the participant received, from their eye movement recording over the text:

$$g(T, E_{P,T}) \longrightarrow \hat{q}_P$$

## 4 MODELS

We introduce discriminative and generative models, presented schematically in Figure 2.

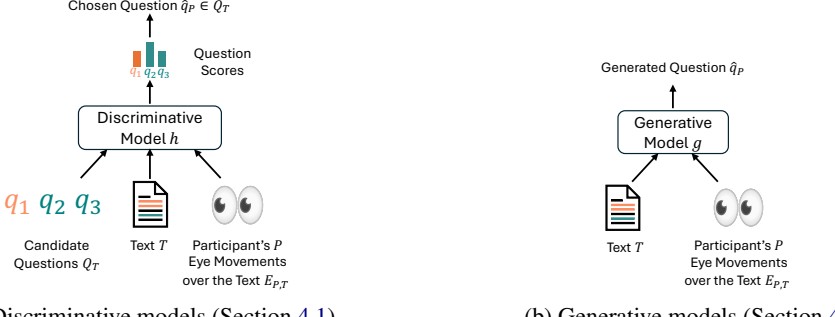

(a) Discriminative models (Section 4.1)     (b) Generative models (Section 4.2)

Figure 2: Two model types for decoding the question presented to a participant before reading the text, from their eye movements over the text. (a) Discriminative models that score candidate questions and are evaluated on question selection accuracy. (b) Generative models that reconstruct the question presented to the reader, and are evaluated on question reconstruction quality.

### 4.1 DISCRIMINATIVE MODELS

READING-TIME INFORMED EMBEDDING SIMILARITY MODELS

We introduce two baseline models which build on the following observations from prior literature: (i) readers tend to spend more time on question-relevant than on question-irrelevant portions of the text (Kaakinen et al., 2002; Malmaud et al., 2020; Hahn & Keller, 2023; Shubi & Berzak, 2023), and (ii) question-relevant text segments tend to have higher semantic similarity to the question compared to question-irrelevant segments (Mitra et al., 2016).

**Question Similarity to RT-Weighted Passage** (Figure 3a) For a participant reading a text of $N$ words, we compute a text embedding, emb$(T)$, as a weighted average of its contextualized word

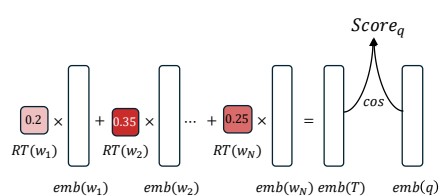 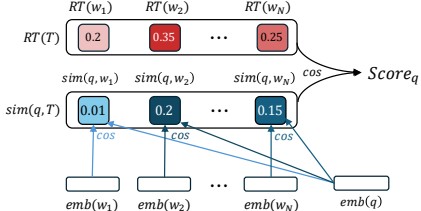

(a) Question Similarity to RT-Weighted Passage    (b) RT Similarity to Question-Word Similarities

Figure 3: Reading-Time Informed Embedding Similarity models for the question selection task.

embeddings, where the weight of each word is the fraction of time spent reading it. Formally: $\text{emb}(T) = \sum_{i=1}^{N} \text{RT}(w_i) \text{emb}(w_i)$. The embeddings for word $w_i$, $\text{emb}(w_i)$, are extracted from RoBERTa (Liu et al., 2019), and $\text{RT}(w_i)$ is the word's Total Fixation Duration (i.e., sum of all the fixation durations) normalized by the text's total reading time. For each candidate question $q_j$, we use its [CLS] token embedding as $\text{emb}(q_j)$, and compute its cosine similarity with $\text{emb}(T)$. We then predict the question with the highest similarity: $\arg\max_{j \in \{1, \ldots, |Q_T|\}} \cos(\text{emb}(T), \text{emb}(q_j))$.

**RT Similarity to Question-Word Similarities** (Figure 3b) An alternative to the model above, compares the vector of word reading times for the text to the vector of word–question similarities. Specifically, for a given candidate question $q_j \in Q_T$ and each text word $w_i$, we first compute $\text{sim}(q_j, w_i) = \cos(\text{emb}(q_j), \text{emb}(w_i))$, where as before, $\text{emb}(w_i)$ is a contextual word embedding from RoBERTa, and $\text{emb}(q_j)$ is the question [CLS] embedding, yielding a length-$N$ vector of similarities $\text{sim}(q_j, T) = [\text{sim}(q_j, w_1), \text{sim}(q_j, w_2), \ldots, \text{sim}(q_j, w_n)]$. We select the question maximizing the cosine similarity between this vector and $\text{RT}(T)$, the length-$N$ vector of speed-normalized Total Fixation Durations for the text words: $\arg\max_{j \in \{1, \ldots, |Q_T|\}} \cos(\text{sim}(q_j, T), \text{RT}(T))$.

STATE-OF-THE-ART NEURAL MODELS

We adapt two state-of-the-art encoder models for eye movements and text. Building on the approach of Radford et al. (2019) for encoding multiple-choice QA items, we use several copies of each text, where instead of combining each copy with a candidate answer, we combine it with a candidate question, and further provide the participant's eye movements over the text $\langle T, E_{P,T}, q \in Q_T \rangle$. The models assign a probability to each triplet and select the highest probability question.

**Haller RNN** (Haller et al., 2022) An RNN model that represents each fixation on the text using eye movement features concatenated with the corresponding RoBERTa-based word embedding. We append an embedding of the question to the input of the model. The model scores each candidate question. Additional details on the model, including a diagram of the modified architecture and the used eye movement features, are presented in Appendix B.1.

**RoBERTEye-Fixations** (Shubi et al., 2024) A transformer-based model which integrates fixation-level eye movement features into RoBERTa. An eye movements feature vector for each fixation is first projected to the dimension of the word embeddings. These fixation vectors are then aligned with their corresponding words via the addition of position embeddings, and finally concatenated with the word sequence. We adapt this model to incorporate candidate question embeddings and modify the classification layer to score questions. Appendix B.2 provides further details, including a model diagram and the eye movement features that the model uses.

## 4.2 GENERATIVE MODELS

We experiment with two approaches for generating the question based on eye movements over the text. The first approach (DalEye-Llama, DalEye-GPT) is fine-tuning, where the task, the text, and the eye movements are provided to the model in the prompt in textual form. The second approach uses the same prompts for zero-shot and few-shot generation from a frontier LLM (Gemini).

**DalEye-Llama** We provide the task, the text, and the eye movement trajectory of a participant over the text to Llama 3.1 (Grattafiori et al., 2024) in *textual* form. The model is fine-tuned to

reconstruct the true question given this input using an autoregressive next-token prediction objective. We use a fixation-based eye movements representation, where each fixation is represented as a tuple containing the index of the fixated word, the word itself, the fixation duration, and the direction of the next saccade. This encoding preserves both the spatial locations and the temporal order of the fixations. This input is preceded by the text and a task description instructing the model to generate the question that was presented to the reader. An example of a prompt is shown in Appendix C.1.

**DalEye-GPT** A similar approach to DalEye-Llama, with identical prompts, where instead we fine-tune a commercial LLM, GPT-4o-mini (Hurst et al., 2024).

**Gemini** We evaluate a frontier commercial LLM, Gemini-3-Pro-Preview (Google DeepMind, 2025) in zero-shot and few-shot settings. In the zero-shot setting, the model receives the same textual input as DalEye-Llama and DalEye-GPT. In the few-shot setting, the prompt further includes 10 randomly drawn trial examples with ground truth questions (see Appendix C.1.2 for details).

In Appendix C.2 we further provide details on DalEye-LLaVA, an alternative, *embedding* based approach to finetuning based on the LLaVA-OneVision language-vision model architecture (Li et al., 2025). Although this model mostly did not perform well on our task, we provide its description and evaluations in the Appendix as a reference point for future embedding based models.

## 5 EXPERIMENTAL SETUP AND EVALUATION

### 5.1 DATA SPLITS

Following prior work on predictive modeling with OneStop (e.g. Shubi et al., 2024), we use a 10-fold cross-validation procedure where each of the 10 data splits has a training, validation, and test set. The OneStop eye tracking data is not i.i.d.; each participant reads multiple paragraphs, and each paragraph is read by multiple participants. To account for this structured nature of the data, the validation set and the test set of each data split are each partitioned into three disjoint parts that capture three different levels of model generalization, ordered by task difficulty:

**New Participant:** Training eye tracking data is available for the text (with each of the three possible questions) but not for the participant.

**New Text:** Training eye tracking data is available for the participant but not for the text.

**New Text & Participant:** Neither the participant nor the text was in the training data.

Each data split includes 64% of the trials in the training set, 17% in the validation set, and 19% in the test set – divided into 9% New Participant, 9% New Text, and 1% New Text & Participant. When aggregated across the 10 splits, 90% of the trials appear in both New Participant and New Text evaluation regimes, and 10% appear in the New Text & Participant regime. We ensure that in each split into training, validation, and test sets, there is a balanced distribution of question types, i.e. $q_{1,c_1}, q_{2,c_2}, q_{3,c_2}$ for each text in each part of the split. Hyperparameter optimization and model selection are performed separately for each split. We assume that the evaluation regime with respect to novelty of texts and participants at test time is *unknown*, and therefore model selection is performed using the entire validation set within each split. Further details on the hyperparameter search space, training, and evaluation procedures are provided in Appendix D.

### 5.2 GOAL SELECTION

As described in Section 4, the discriminative models assign a probability to a triplet $\langle T, E_{P,T}, q \in Q_T \rangle$, and select the candidate question with the highest probability. We report a three-way selection accuracy **All**, as well as accuracy for two specific cases of interest:

**Different critical spans** This evaluation tests the ability to distinguish between questions with disjoint critical spans. We pose the selection problem as a binary decision between $q_{1,c_1}$ and $\{q_{2,c_2}, q_{3,c_2}\}$. Random choice accuracy in this evaluation is approximately 55%, rather than 50%, due to a Monty Hall-type setup arising from grouping two questions under one critical span (see Appendix E).

**Same critical span** This is a more challenging evaluation for differentiating between the two questions with the same critical span $c_2$: $q_{2,c_2}$ versus $q_{3,c_2}$. Note that here we disregard the question whose critical span is $c_1$. The corresponding chance accuracy is 50%.

## 5.3 GOAL RECONSTRUCTION

The generative models are evaluated on the quality of the reconstructed questions. As with other generation tasks, evaluating reconstruction quality is a major challenge. Here, we use several heuristic evaluations for both the surface form and the semantic content of the generated questions.

**Question word accuracy** We test whether the generated question word matches the question word of the ground truth question. The possible question words are What, When, Where, Who, Whom, Which, Whose, Why, and How. An additional category is Other, reserved for all other words starting a question (7% of the questions).

**UIUC question category accuracy** We check whether the generated question has the same semantic category as the ground truth question according to the question-type taxonomy of Li & Roth (2002). We use GPT-4o (Hurst et al., 2024) to categorize the questions into the main taxonomy categories: *entity*, *human*, *numeric*, and *location*, and the subcategories of 'description': *manner*, *reasoning*, *definition*, and *description*. Appendix F.1 includes further details on the classification process and on the agreement of GPT-4o with manual human annotations.

**BLEU** The BLEU surface similarity score (Papineni et al., 2002) between the generated question and the true question.

**BERTScore** (Zhang et al., 2020) A cosine similarity score between the generated question and the true question based on contextual embeddings extracted from RoBERTa.

**QA accuracy** A downstream evaluation with the following logic: the closer the generated question is to the true question, the easier it should be to choose the correct answer for the true question given the reconstructed question. We consider a question as a valid reconstruction of the true question if and only if, given this question, the text, and the four answers for the true question, the model selects the correct answer. We implement this evaluation using RoBERTa fine-tuned for multiple choice QA on RACE (Lai et al., 2017), on the 86% of OneStopQA questions that RoBERTa answers correctly.

### BASELINES

To facilitate model performance interpretation, we introduce the following baseline questions which are composed without eye movement information.

**Human incorrect questions** The two additional human-composed questions for each text, one with a different and one with the same critical span as the true question (Section 2). These questions are guaranteed to fulfill two key criteria: they are well-formed and can be answered from the passage.

**LLM arbitrary questions** An arbitrary question for each passage generated with the most recent frontier LLM, Gemini-3-Pro-Preview, using the prompt presented in Appendix F.2.

**LLM questions from text-only question decoding models** Questions from Llama 3.1 and GPT-4o-mini finetuned to decode the question in using the same experimental setup as DalEye-Llama and DalEye-GPT, but *without* eye movements data.

The questions generated from the LLM baselines enable quantifying the contribution of eye movement information in each of our generative models, above and beyond textual heuristics. This includes effects of possible model exposure to the textual materials of OneStopQA during pretraining.

## 6 RESULTS

### 6.1 GOAL SELECTION

Table 1 presents the aggregated test accuracy across the 10 data splits for the three-way All goal selection, and a breakdown into questions with Different Spans, and the Same Span evaluations. A division of the results into the New Participant, New Text, and New Text & Participant regimes is

Table 1: *Goal selection* test accuracy aggregated over 10 data splits, with 95% confidence intervals. A majority choice is identical to chance because the data is balanced across questions. As the data is not i.i.d. (each participant reads multiple paragraphs, and each paragraph is read by multiple participants), we test for differences between models using a linear mixed effects model with random intercepts and slopes for participants and paragraphs: $is\_correct \sim model + (model \mid participant) + (model \mid paragraph)$. Significant gains over chance performance are marked with '*' $p < 0.05$, '**' $p < 0.01$, and '***' $p < 0.001$. The best performing model in each evaluation is marked in bold. Significant drops compared to the best model are marked with '+++' $p < 0.001$.

| Model | All | Different Spans | Same Span |
|---|---|---|---|
| Chance / Majority Baseline | $33.0 \pm 0.4_{+++}$ | $55.3 \pm 0.4_{+++}$ | $49.9 \pm 0.4_{+++}$ |
| Question Similarity to RT-Weighted Passage | $33.4 \pm 0.3_{+++}$ | $55.1 \pm 3.7_{+++}$ | $50.0 \pm 0.4_{+++}$ |
| RT Similarity to Question-Word Similarities | $34.2 \pm 0.4^{*}_{+++}$ | $54.8 \pm 1.4_{+++}$ | $51.1 \pm 0.5_{+++}$ |
| Haller RNN (Haller et al., 2022) | $41.8 \pm 0.6^{***}_{+++}$ | $65.6 \pm 0.5^{***}_{+++}$ | $52.1 \pm 0.5^{**}_{+++}$ |
| RoBERTEye-Fixations (Shubi et al., 2024) | $\mathbf{49.3 \pm 0.3^{***}}$ | $\mathbf{70.9 \pm 0.5^{***}}$ | $\mathbf{57.3 \pm 0.5^{***}}$ |

reported in Tables 5 to 7 in Appendix G.1.1. Validation set results are reported in Tables 8 to 11 in Appendix G.1.1. A secondary evaluation of the open generative models DalEye-Llama and DalEye-LLaVA on the question selection task is reported in Appendix G.1.2.

Both Reading-Time Informed Embedding Similarity baseline models perform at chance level across all three evaluation regimes, with the exception of RT Similarity to Question–Word Similarities in the *All* condition, where performance is marginally above chance. A possible reason for this outcome lies in the inherent noise in the distribution of reading times of a single participant over a single paragraph, which deems heuristic methods that rely only on this distribution to be ineffective. Haller RNN and RoBERTEye-Fixations on the other hand perform well above chance.

RoBERTEye-Fixations achieves the highest accuracy across all three evaluation regimes. In Table 5 of Appendix G.1.1, we observe that this performance advantage is consistent across the New Participant, New Text, and New Text & Participant evaluations. Thus, the model is able to generalize not only to new participants but importantly also to new texts. Notably, in addition to achieving 70.9% accuracy on the Different Spans evaluation, it is also the only model that is well above chance in the Same Span regime with 57.3% accuracy. While these results confirm that distinguishing between questions that ask about the same portion of the text is indeed much more challenging than distinguishing between questions over different parts of the text, the above-chance performance of RoBERTEye-Fixations on the Same Span evaluation speaks to the fine-grained nature of the information that can be extracted about the reader's information seeking goals from their eye movements alone. Appendix G.1.3 reports a feature ablation analysis of RoBERTEye-Fixations, which suggests that the model relies much more heavily on the ordering of word embeddings according to the sequence of fixations than on any additional features of each individual fixation.

### LINKING MODEL PERFORMANCE TO COGNITIVE INDICATORS OF INFORMATION SEEKING

To tie automated goal decoding to cognitive aspects of gaze behavior during information seeking, we present a model performance analysis of RoBERTEye-Fixations, the top performing model in the goal selection task, based on the framework proposed by Shubi et al. (2025). The analysis examines how features of the experimental trial that are pertinent to cognitive theory of information seeking (e.g. Kaakinen et al. (2002); Hahn & Keller (2023); Shubi & Berzak (2023)) and are not provided to the model, affect model performance. Specifically, it uses the following linear mixed effects model which accounts for the non i.i.d structure of the data: $P(is\_correct) \sim feat_1 + \cdots + feat_n + (1 \mid participant) + (1 \mid paragraph)$, where $P(is\_correct)$ is the probability assigned by RoBERTEye-Fixations to the correct question given an input of a paragraph and eye movements. We use the 10 trial features from Shubi et al. (2025) which capture aspects of the participant's interaction with the textual item, and the item itself, and further include the lexical overlap between the question and critical span, quantified with ROUGE-1-precision. Figure 4 depicts the resulting model coefficients.

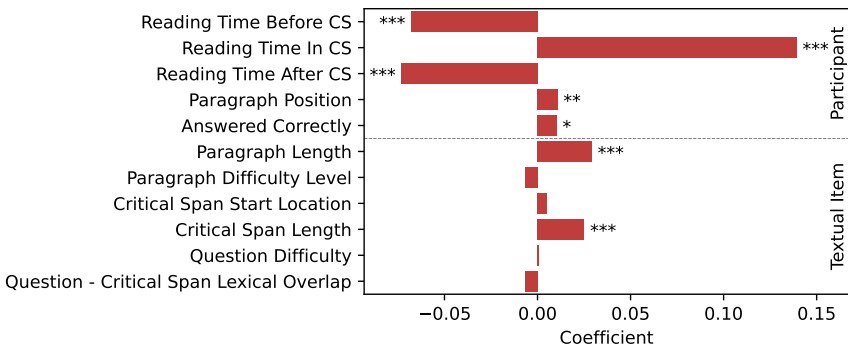

Figure 4: Coefficients from a mixed-effects model which predicts the probability assigned by RoBERTEye-Fixations to the correct question from 11 features of the experimental trial. Predictors are z-normalized, to make the coefficient magnitudes comparable. Coefficient statistical significance after a 11x Bonferroni correction is marked with '*' $p < 0.05$, '**' $p < 0.01$, and '***' $p < 0.001$.

**Participant features over the textual item** Prior studies in psycholinguistics have observed shorter reading times on goal irrelevant compared to goal relevant information, which was linked to rational processing during information seeking (Kaakinen et al., 2002; Hahn & Keller, 2023; Shubi & Berzak, 2023). Intriguingly, we find that the extent to which this holds affects model performance in a highly interpretable manner: longer reading times (per word Total Fixation Duration) before and after the critical span hurt model performance ($p < 10^{-63}$ and $p < 10^{-71}$ respectively), while longer reading times within the critical span improve it ($p < 10^{-275}$). In other words, the more goal directed and resource efficient the reader is, the easier it is to identify their goal automatically. The paragraph's position in the experiment (1-54) is also positively correlated with model performance ($p < 10^{-3}$), likely due to participants' adaptation to the task over time (e.g. Wells et al., 2009; Chromỳ & Tomaschek, 2024). Better reading comprehension is similarly positively associated with model performance ($p < 0.01$), in line with work linking better time division between goal relevant and irrelevant information and reading comprehension performance (Shubi & Berzak, 2023).

**Textual item features** Longer paragraphs are associated with improved model performance ($p < 10^{-6}$), possibly as they allow better separation between question relevant and question irrelevant material, as well as between critical spans of different questions. Longer critical spans, which potentially support more robustness in the extraction of critical span related eye movements, are also positively correlated with model performance ($p < 10^{-8}$). Interestingly, this result complements an opposite trend for models that are trained to distinguish between information seeking and reading for comprehension without receiving the question ahead of time (Shubi et al., 2025), where longer critical spans make it harder for models to distinguish these two reading regimes. Similarly to the results in Shubi et al. (2025), paragraph difficulty level (original or simplified), and critical span location do not have a significant effect on the model's performance. Finally, we do not find evidence for an influence of the lexical overlap of the critical span with the question. Overall, these findings provide converging evidence and expand prior observations from the psycholinguistic literature on information seeking using the developed model as an analytical tool.

## 6.2 GOAL RECONSTRUCTION

Figure 5 presents the goal reconstruction evaluations in the New Participant and the New Text evaluation regimes. Tables 14 and 15 in Appendix G.2 present these results numerically, and also include the New Text & Participant regime, which yields similar results to the New Text regime. In the New Participant evaluation, DalEye-Llama, DalEye-GPT and Gemini few-shot generated questions outperform all the baseline questions on all five evaluation measures ($p < 0.001$ in all cases)[2], except for Gemini few-shot in the Question Category evaluation. These performance gains suggest that these models generalize relatively well to new participants when the text and possible questions appeared in the training set. Gemini few-shot is the top performer in this regime, with the exception of Question Category where DalEye-Llama performs best. Notably, the text-only version

---

[2]Performance differences are tested using: $measure \sim model + (1 \mid participant) + (1 \mid paragraph)$.

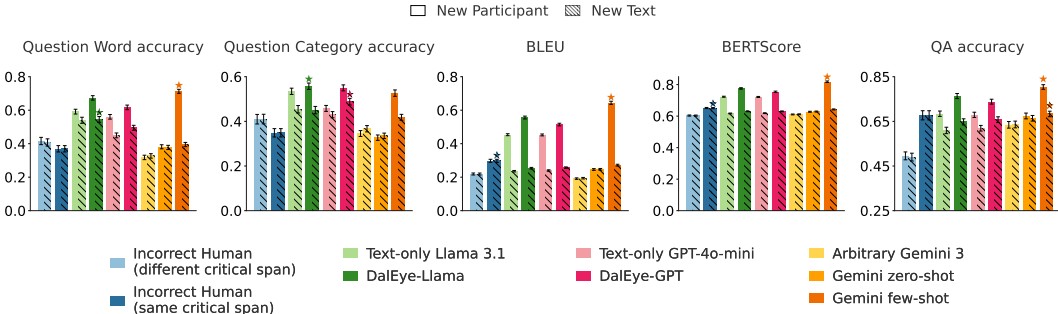

Figure 5: *Goal reconstruction* from eye movements evaluations of the finetuned models DalEye-Llama and DalEye-GPT, as well as zero-shot and few-shot Gemini for (1) the identity of the generated question word, (2) UIUC semantic category of the question, (3) BLEU score, (4) BERTScore, (5) downstream QA accuracy based on the answer selection of a multiple-choice QA model. The models are benchmarked against five baselines that *do not include eye movements*. Two baselines are human-composed questions for a different and the same critical span. Three additional baselines are LLM-generated questions: an arbitrary question about the text generated with Gemini 3, and questions from text-only variants of DalEye-Llama and DalEye-GPT finetuned on the question decoding task using only the text. Results are aggregated over 10 data splits. Presented are intercept coefficients with 95% confidence intervals from linear mixed effects models with random intercepts for participants and paragraphs: $measure \sim 1 + (1 \mid participant) + (1 \mid paragraph)$ for eye movement models and $measure \sim 1 + (1 \mid paragraph)$ for the baselines. The highest score in each combination of evaluation measure and evaluation regime is marked with $\star$.

of DalEye-Llama, Llama 3.1 finetuned for question reconstruction without eye movements, is the strongest baseline for this evaluation regime across all five evaluation measures.

The performance of DalEye-Llama, DalEye-GPT, and Gemini few-shot drops considerably in the New Text regime, in absolute terms and relative to the baselines, suggesting weaker generalization to new texts and questions. However, even in this challenging evaluation regime, all three models consistently outperform their corresponding text-only or arbitrary counterparts, except for DalEye-Llama on the Question Word and Question Category evaluation. This indicates that eye movements are valuable for the generation task even when the textual item is novel. The best performing model differs by evaluation measure: DalEye-Llama on Question Word, DalEye-GPT on Question Category, and Gemini few-shot on QA accuracy. The Incorrect Human (same critical span) questions achieve the best results on BLEU and BERTScore, likely due to lexical overlap and semantic proximity to the correct question, stemming from sharing the same critical span. The performance of Gemini zero-shot is similar across both regimes, because similarly to the human baselines, this model is not finetuned, and it tends to perform poorly in both regimes.

Table 16 in Appendix G.2 presents examples of questions generated by DalEye-Llama, DalEye-GPT, and Gemini. Table 18 in Appendix G.2 presents experimental results for DalEye-Llama using a differently worded task prompt, which yields similar results, as well as two alternative textual representations of the eye movement sequence, one based on word-level features and another combining word- and fixation-level features. The alternative representations yield inferior results to those presented above.

# 7    RELATED WORK

**Analyses of Goal Based Reading** While most prior work in psycholinguistics focuses on reading for comprehension (Rayner et al., 2012), goal or task based reading has been acknowledged as a key frontier in this area (Radach & Kennedy, 2004). Several studies found differences in eye movements between reading for comprehension and tasks that define a reading manner: skimming, speed reading and proofreading (Just et al., 1982; Kaakinen & Hyönä, 2010; Schotter et al., 2014; Strukelj & Niehorster, 2018; Chen et al., 2023). Additional studies found differences between reading for comprehension and tasks of target word search (Rayner & Raney, 1996; Rayner & Fischer, 1996), word verification (Radach et al., 2008), and topic search (White et al., 2015). Prior work also analyzed

eye movements during linguistic annotation tasks (Tomanek et al., 2010; Tokunaga et al., 2017). Differences in reading patterns were also found when readers adopt different perspectives on a text (Kaakinen et al., 2002; Kaakinen & Hyönä, 2008), or given different learning goals (Rothkopf & Billington, 1979). Overall, these works consistently found a systematic influence of the reading task on reading patterns. We build most directly on Kaakinen et al. (2015), Malmaud et al. (2020), Hahn & Keller (2023), and Shubi & Berzak (2023), who compared eye movements in reading for comprehension and *open-ended information seeking*. These studies showed task conditioning effects on reading patterns, especially as a function of the task-relevance of the textual information. Similarly to these studies, we formulate the information seeking task as a reading comprehension question.

**Decoding Cognitive State from Eye Movements** Multimodal modeling of eye movements and text has been a growing research area in recent years (Reich et al., 2025), with applications ranging from improving NLP (e.g. Klerke et al., 2016; Mishra et al., 2016; Barrett & Hollenstein, 2020; Sood et al., 2020; Khurana et al., 2023; Deng et al., 2024; López-Cardona et al., 2025) to prediction of eye movements (e.g. Hollenstein et al., 2021a; Bolliger et al., 2023; Deng et al., 2023; Chen et al., 2024). Within this line of research, several studies focused on *cognitive state decoding*: prediction of language-related aspects of the reader and their cognitive state from eye movements. These include studies on prediction of the reader's linguistic background (Berzak et al., 2017; Reich et al., 2022; Skerath et al., 2023), language proficiency (Berzak et al., 2018), reading comprehension (Ahn et al., 2020; Reich et al., 2022; Shubi et al., 2024), subjective readability (Reich et al., 2022) and perceived relevance (Bhattacharya et al., 2020a;b).

**Decoding Reading Goals** Most notably, two studies addressed decoding of the reader's goals. Hollenstein et al. (2021b) classified reading for comprehension versus annotation of semantic relations, Shubi et al. (2025) classified reading for comprehension versus information seeking. Importantly, in both studies, the decoding tasks are *procedural*, where the goal is to distinguish between a small set of *pre-defined* manners of reading that are not specific to a particular text. In contrast, our study addresses a related but conceptually and practically different task, which is *semantic*; instead of a small number of categories that can apply to any text, we have hundreds of text-specific goals that can be *arbitrary* in nature. See further discussion on the procedural versus semantic distinction in Shubi & Berzak (2023). Furthermore, these and other studies on cognitive state decoding typically use *discriminative*, encoder-based models that are limited to classification tasks. Here, we also develop *generative*, decoder-based language models. We envision that such models will have broad relevance for future eye movement conditioned text generation tasks.

## 8 SUMMARY AND DISCUSSION

We introduce a new challenge of both scientific and practical value: decoding arbitrary information seeking goals from eye movements in reading. We tackle this challenge in two formulations, goal selection and goal reconstruction, using a number of evaluation frameworks and modeling approaches. The best performing model on the selection task is able to predict the correct question with a considerable degree of success, even when the candidate reading goals pertain to the same information in the text. The performance of the model is driven by cognitively interpretable factors of human information seeking. Our results further suggest that although the reconstruction task is extremely challenging, meaningful progress can also be made in this open-ended regime. Overall, we find that effective modeling can extract highly valuable information regarding specific information seeking goals at the challenging granularity level of a single paragraph.

Automatic decoding of information seeking goals can pave the way for new applications with positive societal value. In the future, media outlets and providers of municipal and governmental services could better understand the information needs of users who access their websites and render information in a manner that better meets these needs. Automatic identification of information seeking goals can also be used for real-time assistance to special populations, when accessing critical information on the web. Future e-learning systems could assess students' progress on information-seeking tasks and monitor their information-seeking skills over time. Finally, one can imagine interactive reading interfaces that include real-time text personalization, such as simplification and suggestion of additional relevant information, according to the reader's information seeking goals. The present study is a first step in enabling such applications.

## 9 ETHICAL CONSIDERATIONS

The OneStop Eye Movements dataset used in this work was collected by Berzak et al. (2025a). The study was conducted under an institutional IRB protocol, and all the participants provided written consent before participation. The data is anonymized. Analyses and predictive modeling of task-based reading are among the primary use cases for which the data was collected.

While the current work is a scientific proof of concept and is not aimed at a particular user-facing application, one has to consider potential use cases and risks for the presented task. In particular, it is important to note that given the current level of accuracy of goal decoding, educational and assistive technologies that rely on this task may be unreliable and introduce biases for various individuals and groups, such as L2 readers and groups with reading and cognitive impairments. Additional data collection and analyses are needed to assess such biases.

Prior work has demonstrated that eye movements can be used for user identification (e.g. Bednarik et al., 2005; Jäger et al., 2020). We do not perform user identification in this study. We further emphasize that future reading goal decoding technologies must be used in real-world applications only with explicit consent from potential users to have their eye movements collected and analyzed for this purpose.

## 10 ACKNOWLEDGMENTS

This work was supported by ISF grant 1499/22.

## 11 REPRODUCIBILITY STATEMENT

We describe the model training and selection procedure, the evaluation protocol, the hyperparameter search space for each model, and the hardware and software specifications in Section 5 and appendix D. The OneStop dataset (Berzak et al., 2025a) used in the experiments and described in Section 2 is publicly available at https://osf.io/2prdq/. The code for the paper, which implements all the models, experimental procedures, and analyses, is available at: https://github.com/lacclab/Open-Ended-Goal-Decoding.

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

APPENDIX

## A  N-GRAM OVERLAP BETWEEN QUESTIONS AND TEXTS

To assess the extent to which the question answering task in OneStop (Berzak et al., 2025a) could be solved through simple lexical matching, we computed the n-gram overlap between each target question and its corresponding passage in OneStopQA (Berzak et al., 2020). We measured this overlap using the proportion of matching unigrams (ROUGE-1[3]), bigrams (ROUGE-2), and unigrams restricted to content words, with respect to the number of n-grams in the question (precision), in the text (recall), and the harmonic mean of the two ($F_1$). We further provide a breakdown of the question overlap with the passage into the words within and outside the question's critical portion of the text. We compared this overlap with the corresponding overlap in SQuAD (Rajpurkar et al., 2016), an extractive QA dataset in which answers appear verbatim in the text. The results can be seen at table 2.

Table 2: Average overlap results for OneStop and SQuAD, comparing questions to different parts of the context. Values in brackets refer to the analysis using only content words from both the question and the context. P (precision), R (recall), and $F_1$ refer to overlap proportion with respect to question length, context length, and their harmonic mean, respectively.

| Dataset | Text Part | ROUGE-1 | | | ROUGE-2 | | |
|---|---|---|---|---|---|---|---|
| | | P | R | $F_1$ | P | R | $F_1$ |
| SQuAD | Paragraph | 0.527 (0.488) | 0.071 (0.052) | 0.122 (0.091) | 0.208 | 0.021 | 0.037 |
| OneStop | Paragraph | 0.463 (0.376) | 0.059 (0.040) | 0.104 (0.071) | 0.127 | 0.012 | 0.022 |
| | CS | 0.338 (0.285) | 0.126 (0.097) | 0.176 (0.137) | 0.100 | 0.033 | 0.048 |
| | out of CS | 0.283 (0.153) | 0.050 (0.024) | 0.084 (0.040) | 0.034 | 0.005 | 0.008 |

We found that the unigram overlap between OneStop questions and passages (46%) is lower than the corresponding overlap in SQuAD (53%). We similarly find a smaller overlap in OneStop for content words (38% OneStop, 49% SQuAD), and for bigrams (12% OneStop, 20% SQuAD). We note that these comparisons are adequate with the ROUGE measures as the OneStop and SQuAD datasets have similar passage lengths on average (110 OneStop, 120 SQuAD) and question lengths (10 words on average in both). We further found that the n-gram overlap of the questions in OneStop with the critical span text is relatively low (33% for unigrams, 10% bigrams), and only slightly higher than with the text outside of the critical span (28% unigrams, 3% bigrams). This further suggests that the relevant text does not differ drastically from non-relevant text in terms of substring overlap with the question. Overall, we find that the overlap in OneStopQA is relatively low, both in absolute terms and compared to SQuAD, providing empirical support for the inferential nature of the information seeking tasks in OneStop.

## B  DISCRIMINATIVE MODELS

The figures and descriptions below, depicting model architectures, show the processing of a single candidate question. As mentioned in Section 4, each candidate question is processed independently.

### B.1  HALLER RNN

Given a text (a paragraph in our case) $T$ and a candidate question $q \in Q_T$, the model suggested by Haller et al. (2022) and adapted by us performs the following steps:

1. **Word Embeddings**: LLM-based contextualized word embeddings, for $T$ and $q$, resulting in three subsequences: (1) $Z_{[CLS]}$, (2) $[Z_{q_{t_1}}, \ldots, Z_{q_{t_l}}]$, and (3) $[Z_{T_{t_1}}, \ldots, Z_{T_{t_x}}]$.

---

[3]We use ROUGE metrics (Lin, 2004), implemented by the `rouge` python package, version 1.0.1.

2. **Fixation Sequence**: Paragraph-only token-level embeddings are sum-pooled to word-level embeddings, and ordered by the fixation sequence, including repetitions of words according to the number of fixations on that word. Then, each embedding in the sequence is concatenated with the respective fixation features. The fixation-level features are:

   - Horizontal position of the fixation
   - Total gaze duration (sum of all fixations on the word; repeated for each fixation on that word)
   - Duration of first fixation
   - Duration of outgoing saccade
   - Horizontal distance of outgoing saccade
   - Vertical distance of outgoing saccade
   - Total distance of outgoing saccade
   - Duration of incoming saccade
   - Horizontal distance of incoming saccade
   - Vertical distance of incoming saccade

3. **Projection**: $Z_{[CLS]}$ and $[Z_{q_{t_1}}, \ldots, Z_{q_{t_l}}]$ are projected to a higher dimension to fit the dimension of the concatenated embeddings from the previous step.

4. **RNN Input Sequence**: The processed embeddings are then concatenated one after the other to create the RNN input sequence.

5. **Classifier**: Finally, a classifier layer scores each candidate question.

Figure 6a depicts the architecture adapted to our task.

### B.2 RoBERTEye-Fixations

Formally, given a paragraph $T$ and a candidate question $q \in Q_T$, the model constructs two parallel representations:

1. **Textual Representation** ($Z_W$): The input follows the format $Z_W = [\text{CLS}; T; \text{SEP}; q; \text{SEP}]$.

2. **Fixation-Level Eye Movement Representation** ($Z_{E_T}$): The eye movement sequence is defined as $Z_{E_T} = [Z_{E_{f_1}}, ..., Z_{E_{f_m}}]$. The fixation representation is computed as:

$$Z_{E_{f_i}} = \text{FC}(E_{f_i}) + \text{Emb}_{\text{pos}}(w_i) + \text{Emb}_{\text{eye}}$$

   Here, $E_{f_i}$ captures fixation properties (e.g., duration, position) for fixation $f_i$ on word $w_i$. The fully connected layer FC projects this feature vector into the word embedding space, $\text{Emb}_{\text{pos}}(w_i)$ is the positional embedding of $w_i$ used to map each fixation to the corresponding word, and $\text{Emb}_{\text{eye}}$ is a learnable embedding marking the presence of eye movement information. The fixation-level features are described in Table 3.

3. **Fusion and Prediction**: The combined sequence $[Z_{E_T}; \text{SEP}_E; Z_W]$ is processed by the transformer encoder, where $\text{SEP}_E$ is a separator token initialized the same as SEP. The final CLS token representation is passed through two fully connected layers to score each candidate question.

Figure 6b depicts the adapted architecture.

Table 3: Eye movement and word property features used by RoBERTEye-Fixations and DalEye-LLaVA. See Berzak et al. (2025a) for further details.

| Feature Name | Description |
|---|---|
| **Word-Level Eye Movement Features** | |
| IA_DWELL_TIME | The sum of the duration across all fixations that fell in the current interest area |
| IA_DWELL_TIME_% | Percentage of trial time spent on the current interest area. |
| IA_FIXATION_% | Percentage of all fixations in a trial falling in the current interest area. |
| IA_FIXATION_COUNT | Total number of fixations falling in the interest area. |
| IA_REGRESSION_IN_COUNT | Number of times interest area was entered from a higher IA_ID. |
| IA_REGRESSION_OUT_FULL_COUNT | Number of times interest area was exited to a lower IA_ID. |
| IA_RUN_COUNT | Number of times the Interest Area was entered and left (runs). |
| IA_FIRST_FIX_PROGRESSIVE | Checks whether the first fixation in the interest area is a first-pass fixation. |
| IA_FIRST_FIXATION_DURATION | Duration of the first fixation event that was within the current interest area |
| IA_FIRST_FIXATION_VISITED_IA_COUNT | Number of distinct interest areas visited before the first fixation on the current area. |
| IA_FIRST_RUN_DWELL_TIME | Dwell time of the first run. |
| IA_FIRST_RUN_FIXATION_COUNT | Number of all fixations in a trial falling in the first run of the current interest area. |
| IA_SKIP | IA_SKIP = 1 if the area received no fixation in first-pass reading. |
| IA_REGRESSION_PATH_DURATION | Total fixation duration on the current interest area until moving to a higher IA_ID. |
| IA_REGRESSION_OUT_COUNT | Exits from current IA to lower IDs before fixating a higher ID. |
| IA_SELECTIVE_REGRESSION_ PATH_DURATION | Duration of fixations and refixations on the current interest area until eyes move to a higher ID. |
| IA_LAST_FIXATION_DURATION | Duration of the last fixation event that was within the current interest area. |
| IA_LAST_RUN_DWELL_TIME | Dwell time of the last run. |
| IA_LAST_RUN_FIXATION_COUNT | Number of fixations during the last run (sequence) within the current interest area. |
| IA_TOP | Y coordinate of the top of the interest area. |
| IA_LEFT | X coordinate of the left-most part of the interest area. |
| IA_FIRST_FIX_PROGRESSIVE | Whether the first fixation in the interest area is progressive (left-to-right, first-pass). |
| normalized_Word_ID | Position in the paragraph of the word interest area, normalized from zero to one. |
| total_skip | Binary indicator whether the word was fixated on. |
| **Paragraph-level Features** | |
| PARAGRAPH_RT | Reading time of the entire paragraph. |
| **Fixation-level Fixation Features** | |
| CURRENT_FIX_INDEX | The position of the current fixation in the trial. |
| CURRENT_FIX_DURATION | Duration of the current fixation. |
| CURRENT_FIX_PUPIL | Average pupil size during the current fixation. |
| CURRENT_FIX_X | X coordinate of the current fixation. |
| CURRENT_FIX_Y | Y coordinate of the current fixation. |
| **Fixation-level Saccade Features** | |
| NEXT_FIX_ANGLE, PREVIOUS_FIX_ANGLE | Angle between the horizontal and the line from the current to the next/previous fixation. |
| NEXT/PREVIOUS_FIX_DISTANCE | Distance between current fixation and next/previous fixation. |
| NEXT_SAC_AMPLITUDE | Amplitude of the following saccade in degrees of visual angle. |
| NEXT_SAC_ANGLE | Angle between the horizontal plane and the direction of the next saccade. |
| NEXT_SAC_AVG_VELOCITY | Average velocity of the next saccade. |
| NEXT_SAC_DURATION | Duration of the next saccade in milliseconds. |
| NEXT_SAC_PEAK_VELOCITY | Peak values of gaze velocity (in visual degrees per second) of the next saccade. |
| NEXT_FIX_INTEREST_AREA_INDEX | Index of the interest area where the next fixation lands |
| **Word Properties** | |
| gpt2_surprisal | Difficulty predicted by GPT-2 model |
| wordfreq_frequency | Frequency of word in language usage |
| word_length | Number of characters in the word |
| start_of_line | Indicates word is at start of line |
| end_of_line | Indicates word is at end of line |
| is_content_word | Is the word a content word (noun, verb, etc.) |
| left_dependents_count | Count of dependents left of word |
| right_dependents_count | Count of dependents right of word |
| distance_to_head | Syntactic distance to head word |

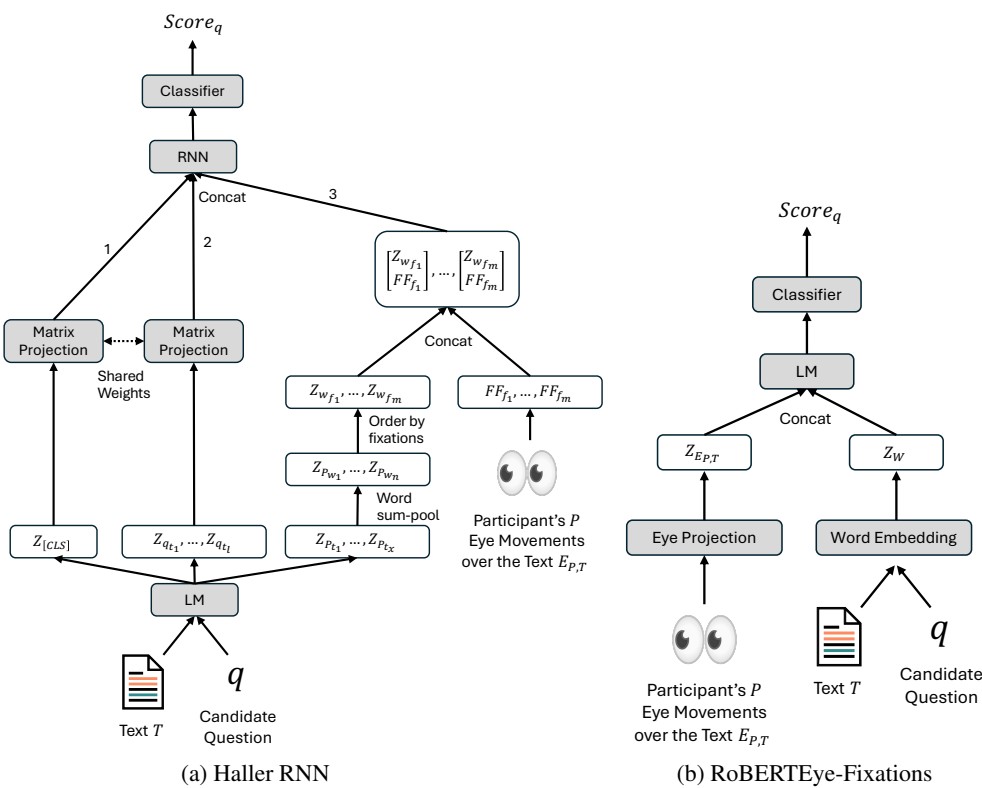

Figure 6: Visualization of the Haller RNN and RoBERTEye-Fixations architectures. $LM$ stands for a language model, $RNN$ for a recurrent neural network. $FF_{f_i}$ stands for the fixation features and $w_{f_i}$ for the word corresponding to the $i$-th fixation.

## C GENERATIVE MODELS

### C.1 PROMPTS FOR DALEYE-LLAMA, DALEYE-GPT AND GEMINI

In Appendix C.1.1, we present the task prompt and eye-movement input representations used for question generation with DalEye-Llama, DalEye-GPT, and Gemini. Variants of these prompts and inputs are described in Appendix G.2.3. The few-shot examples selection method is described in Appendix C.1.2.

### C.1.1 INPUT FORMAT

```
"""
You will be given data from an eye-tracking for reading experiment
in which participants first read a question about a paragraph,
then read the paragraph, and finally answered the question.

Input: You will be provided with a paragraph and eye movements of
a single participant over that paragraph.
Output: Your task is to generate the question that was presented
to the participant prior to reading the paragraph.

Eye Movements Representation:
You will receive the eye movements data for the paragraph
formatted as {FORMAT}

Instructions:
Output only the original question presented to the reader,
matching it as best as you can. DO NOT include any additional
commentary or explanation.

{PARAGRAPH}
{SCANPATH}
"""
```

{PARAGRAPH} is the textual paragraph, for example: `"The quick brown fox jumps..."`.

{SCANPATH} is the eye movements data which depends on the {FORMAT} depicted below.

### C.1.2 GEMINI FEW-SHOT INPUT

In the few-shot setting, we prepend 10 sampled input–output examples, each pairing an eye movement trajectory with its corresponding paragraph and question. Examples are selected to match the evaluation regime while preventing information leakage: for *New Text*, the examples come from the same participant but different articles; for *New Participant*, from the same paragraph but different participants; and for *New Participant & Text*, from different participants and different articles. The examples precede the test instance in the prompt as shown below.

```
"""
    {SYSTEM_MESSAGE}

    <Example>
        <INPUT>
            Paragraph:
            {PARAGRAPH_EXAMPLE}
            Fixations_data:
            {SCANPATH_EXAMPLE}
        </INPUT>
```

```
    <OUTPUT>
        {EXAMPLE_TRUE_QUESTION}
    </OUTPUT>
</Example>
... # X 10 examples

    Paragraph:
    {PARAGRAPH_TEST}
    Fixations_data:
    {SCANPATH_TEST}
"""
```

{SYSTEM_MESSAGE} is the same input prompt as in section C.1.1 (excluding the paragraph and scanpath data).

The four parameters with name format {PARAGRAPH/SCANPATH_EXAMPLE/TEST} are formatted as the {PARAGRAPH} and {SCANPATH} in section C.1.1 for each few-shot example and the test instance, correspondingly. {EXAMPLE_TRUE_QUESTION} is the corresponding question for each example.

## C.2 DALEYE-LLAVA

This model is based on the LLaVA-OneVision language-vision model architecture (Li et al., 2025). The model input consists of a task prompt, the text, and eye movement feature embeddings for the text words. We keep Qwen 2-0.5B (Yang et al., 2024) as the language model backbone, and replace the vision encoder with a novel trainable eye movement encoder, which encodes eye movement features using fully connected and convolutional layers. The eye movement embeddings are positionally aligned with the text words. The model is fine-tuned using an autoregressive next-token prediction objective for the correct question. Additional details, including a model diagram and eye movement features, are provided below. The model input is structured as an instruction-style prompt comprising three components: (1) a task description, (2) the paragraph text, and (3) the participant's eye movement features, positionally aligned with the paragraph words. The prompt concludes with an <eos> token. The entire sequence is tokenized, and training uses teacher forcing, with cross-entropy loss applied to the ground-truth question tokens.

The forward architecture mirrors LLaVA's design, with three main stages, as depicted in Figure 7:

1. **Text encoding:** The task prompt and paragraph are tokenized and embedded by the backbone language model.

2. **Fixation encoding:** Eye movement features (defined in Table 3) are processed by a dedicated fixation encoder. The encoder first applies a two-layer MLP with LeakyReLU activation and dropout over the feature dimension (Linear $\rightarrow$ LeakyReLU $\rightarrow$ Dropout $\rightarrow$ Linear $\rightarrow$ LeakyReLU) and then applies two temporal 1D convolution layers (kernel size 3, stride 1, padding 1), with LeakyReLU after each convolution and dropout between the two convolutions. This yields a sequence of fixation embeddings.

3. **Fusion:** Fixation embeddings replace the <image> placeholder token in the input sequence. Word tokens are assigned positional IDs according to their order in the paragraph, while each fixation embedding inherits the positional ID of the corresponding fixated word.

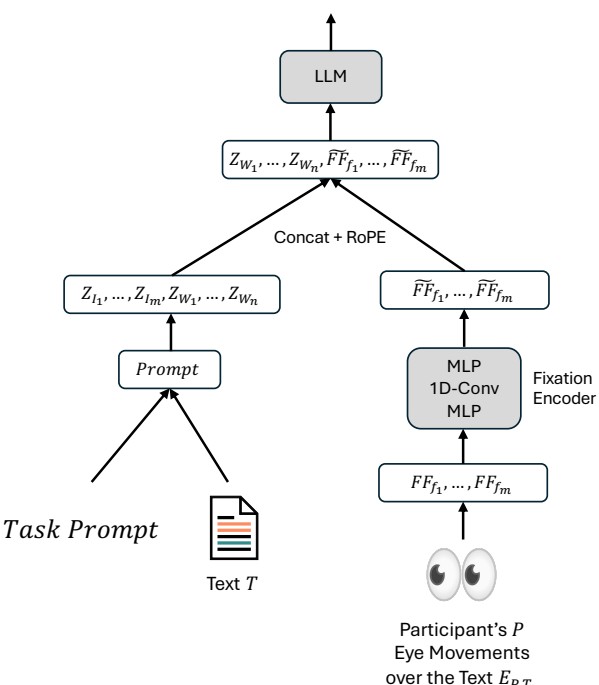

Figure 7: DalEye-LLaVA architecture. Text tokens and eye movement features are fused via a fixation encoder and rotary positional embeddings before being passed to the LLM.

## D    MODEL TRAINING AND HYPERPARAMETERS

We use a stringent evaluation protocol, in which paragraphs are assigned to training, validation, and test sets at the *article level*, such that all the paragraphs from the same article appear in the same portion of each data split.

### D.1    DISCRIMINATIVE MODELS

Since the models we use were developed for different tasks and datasets, we conducted a hyperparameter search for each model. The search space for each model is described below. In all cases, it includes the optimal parameters reported in the work that introduced the model, extended to provide a fair comparison between models.

For all neural models, we train with learning rates of $\{0.00001, 0.00003, 0.0001, 0.0002\}$, following Shubi et al. (2024). Additionally, for all models that make use of word embeddings, we include both frozen and unfrozen language model variants in the search space.

- For **Haller RNN**, we search over LSTM hidden sizes of $\{10, 40, 70, 140\}$, using one layer and a dropout rate of $0.1$. The model is trained with and without freezing language model parameters.
- For **RoBERTEye-Fixations**, we search over dropout rates of $\{0.1, 0.3, 0.5\}$ for the eye movement projection layer. The model is trained with and without freezing language model parameters.

We train the deep-learning-based models for a maximum of 40 epochs, with early stopping after 8 epochs if no improvement in the validation error is observed. A single training epoch took roughly 5 minutes for RoBERTEye-Fixations, 10 minutes for Haller RNN, and 30 minutes for DalEye-Llama and DalEye-LLaVA. Each individual run was capped at 24 hours. Following Liu et al. (2019); Mosbach et al. (2021); Shubi et al. (2024), we use the AdamW optimizer (Loshchilov & Hutter, 2018) with a batch size of 16. RoBERTEye-Fixations uses a linear warm-up ratio of 0.06 and a weight decay of 0.1. We standardize each eye movement feature using statistics computed on the training set, to zero mean unit variance.

Both reading-time informed embedding similarity baseline models use word embeddings from the RoBERTa-Large (Liu et al., 2019) language model.

### D.2    GENERATIVE MODELS

**DalEye-Llama**    This model is fine-tuned using Unsloth with the Meta-Llama-3.1-8B backbone loaded in 4-bit precision. Fine-tuning uses Low-Rank Adaptation (LoRA)(Hu et al., 2022) with all base model parameters frozen. We train only the LoRA adapters, configured with rank $r = 16$, scaling factor $\alpha = 16$, and RS-LoRA regularization. LoRA is applied to the transformer projection modules: q_proj, k_proj, v_proj, up_proj, down_proj, o_proj, and gate_proj. The model is fine-tuned for 2 epochs with a linear scheduler and warm-up of 10 steps, batch size of 1, gradient accumulation over 2 steps, AdamW-8bit optimizer with learning rate $1 \times 10^{-4}$, and weight decay of 0.01.

**DalEye-GPT**    We fine-tune *gpt-4o-mini-2024-07-18* using the OpenAI API for one epoch, with an automatically selected batch size of 4 and a learning-rate multiplier of 1.8.

**DalEye-LLaVA**    The LLaVA backbone remains frozen, and training employs LoRA with dropout rate of 0.1, RS-LoRA, and rank $r = 16$. The hyperparameter search includes learning rates of $1 \times 10^{-5}, 3 \times 10^{-5}, 1 \times 10^{-4}$, fixation embedding hidden sizes of $512, 1024$, and early stopping after 7 epochs without validation improvement.

The discriminative models, DalEye-Llama, and DalEye-LLaVA were trained using the PyTorch Lightning (Falcon & team, 2024) library on NVIDIA A100-40GB and L40S-48GB GPUs.

We utilize the Hugging Face implementation of LLaVA-OneVision, specifically *LLaVA-hf/LLaVA-onevision-qwen2-0.5b-si-hf*. For GPT-4o we use version *gpt-4o-2024-08-06*. For Gemini 3 we use

*gemini-3-pro-preview* version last updated on November 2025. We use Unsloth (version 2025.9.7) (Han et al., 2023) to fine-tune the Llama model. For the QA model we use *LIAMF-USP/roberta-large-finetuned-race*.

There are roughly 355M trainable parameters for RoBERTEye-Fixations, reduced to 3M when the RoBERTa backbone is frozen. Haller RNN consists of 357M trainable parameters, or 2.4M when the backbone is frozen. DalEye-LLaVA has 505M parameters, out of which roughly 12M were unfrozen. DalEye-Llama has 8.1B parameters, out of which roughly 42M were unfrozen.

Statistical testing is done with the Julia MixedModels package (Bates et al., 2022).

# E  MONTY HALL-TYPE SETUP

Table 4 illustrates the structure of the question selection task and the resulting chance accuracies under the three evaluation regimes. In the three-way All setting (a), the model selects among three candidate questions, yielding a chance accuracy of $3/9 = 33.3\%$. In the Different critical spans evaluation (b), success is relaxed to selecting either of the two questions associated with the critical span $c_2$ when it represents the correct span. Here, the probability of a correct random choice increases to $5/9 = 55.5\%$, analogous to a Monty Hall–type setup. Finally, in the Same critical span evaluation (c), the task reduces to a balanced binary choice between the two questions sharing $c_2$, resulting in a chance accuracy of $2/4 = 50\%$.

Table 4: Overview of (a) the three-way question selection task and the breakdown of this task into (b) questions with different critical spans and (c) questions with the same critical span. Chance accuracy under the three evaluation regimes: (a) $3/9$ (33.3%), (b) $5/9$ (55.5%), (c) $2/4$ (50%).

| Predicted | (a) All | | | (b) Different critical spans | | | (c) Same critical span | | |
|---|---|---|---|---|---|---|---|---|---|
| | $q_{1,c_1}$ | $q_{2,c_2}$ | $q_{3,c_2}$ | $q_{1,c_1}$ | $q_{2,c_2}$ | $q_{3,c_2}$ | $q_{1,c_1}$ | $q_{2,c_2}$ | $q_{3,c_2}$ |
| True $q_{1,c_1}$ | ✓ | ✗ | ✗ | ✓ | ✗ | ✗ | — | — | — |
| $q_{2,c_2}$ | ✗ | ✓ | ✗ | ✗ | ✓ | ✓ | — | ✓ | ✗ |
| $q_{3,c_2}$ | ✗ | ✗ | ✓ | ✗ | ✓ | ✓ | — | ✗ | ✓ |

# F  QUESTION RECONSTRUCTION

## F.1  QUESTION ANNOTATION INTO UIUC CATEGORIES

This section provides additional information regarding the question annotation into UIUC categories (Li & Roth, 2002). Specifically, in Appendix F.1.1 we include the full prompt given to the model, in Appendix F.1.2 examples of questions and their corresponding annotations, and in Appendix F.1.3 agreement with human annotators.

### F.1.1  FULL PROMPT

```
You are an expert at classifying questions into the standard UIUC
**main question categories**.

The main categories include:

- ABBR: Abbreviation
For example:
  - What does the abbreviation AIDS stand for?
  - What is the abbreviation for micro?
  - What is the abbreviation of the company name 'General Motors'?
  - What does G.M.T. stand for?

- ENTITY: Entity
For example:
  - What kind of animal is Babar?
  - What killed Bob Marley?
  - What is a fear of weakness?
  - Where does your hair grow the fastest?

- DESCRIPTION: Description
For example:
  - What do Mormons believe?
  - What is the history of skateboarding?
  - What is the difference between a generator and an alternator?
  - Where do rocks come from?

- MANNER: Manner
For example:
  - How do I get another city's newspaper?
  - How do you solve "Rubik's Cube"?
  - How do you look up criminal records on the Internet?
  - How do you find oxidation numbers?

- REASON: Reason
For example:
  - What is the purpose of a car bra?
  - What makes a tornado turn?
  - What causes the redness in your cheeks when you blush?
  - Why do horseshoes bring luck?

- DEFINITION: Definition
For example:
  - What is a dental root canal?
  - What is the contents of proposition 98?
  - Hazmat stands for what?
  - What does the name Billie mean?

- HUMAN: People or groups
```

```
For example:
  - Who invented baseball?
  - What stereo manufacturer is 'Slightly ahead of its time'?
  - Who played the original Charlie's Angels?
  - What company's logo is a 'W' in a circle?

- LOCATION: Geographic locations
For example:
  - Where is the highest point in Japan?
  - What European city do Nicois live in?
  - What is Answers.com 's address?
  - What U.S. state borders Illinois to the north?

- NUMERIC: Numbers, quantities, and dates
For example:
  - What is the temperature for baking Peachy Oat Muffins?
  - How many colleges are in Wyoming?
  - What is the average temperature in the Arctic?
  - What is the speed of light?

---
Return an array of tuples in the format:
```
[ (0, "HUMAN"), (1, "DESCRIPTION"), (2, "NUMERIC"), (3,
"DESCRIPTION"), ]
```
If unsure, choose the closest matching category.
"""
```

### F.1.2 CATEGORIZATION EXAMPLES

For each UIUC question category we present an example question sourced from OneStopQA (Berzak et al., 2020):

- REASON - Why does Myslajek mention Russia, Lithuania and Belarus?
- NUMERIC - Approximately how many taxi drivers are there in the UK?
- MANNER - How does Myslajek react to what he sees between the two paw prints?
- LOCATION - Where was wolf-hunting banned in 1995?
- ENTITY - Which of the following will be featured at Pestival 2013?
- HUMAN - Who threw the bottle into the Baltic Sea?
- DESCRIPTION - What does Angella think of the state of the sea today?

### F.1.3 AGREEMENT WITH HUMAN ANNOTATORS

To evaluate the quality of the GPT-4o question category classifications, we randomly sampled 100 questions. A human annotator (one of the paper's authors) manually labeled them with a UIUC category. We find an 86% agreement (0.794 Cohen's kappa) between the human annotations and the model classifications.

### F.2 ARBITRARY AND TEXT-ONLY QUESTION GENERATION

Below is the prompt used for generating questions using the Text-only Llama 3.1, Text-only GPT-4o-mini, Arbitrary Gemini 3, and Text-only LLaVA-OneVision.

```
Task Description:
Your task is to generate a question a reader had in mind before
reading a given paragraph. The input data is the paragraph itself
in a standard textual format.
```

```
Input Format:
You will receive a paragraph in plain text format:

[Paragraph text]

Expected Output:
Generate the exact original question provided to the reader,
accurately inferred from the paragraph content. Identify key
themes, concepts, or statements in the paragraph that strongly
indicate the question that initially motivated the reading.

Instructions:

Your output must precisely match the original question presented
to the reader.

Focus specifically on central concepts, themes, or statements
within the paragraph.

Produce only the exact original question as your output.
```

# G ADDITIONAL RESULTS

## G.1 GOAL SELECTION

### G.1.1 TEST AND VALIDATION RESULTS ACROSS SUBTASKS AND EVALUATION REGIMES

The tables below present a breakdown of the test and validation results by the All Spans, Different Spans and Same Spans tasks, and by New Text, New Participant and New Text & Participant evaluation regimes aggregated across 10 cross-validation splits.

Model performance is compared to the Majority class baseline using a linear mixed effects model. In R notation: $is\_correct \sim model + (model \mid participant) + (model \mid paragraph)$. Significant gains over this baseline are marked with '*' $p < 0.05$, '**' $p < 0.01$ and '***' $p < 0.001$ in superscript. The best performing model in each evaluation regime is marked in bold. Significant drops compared to the best model are marked in subscript with '+' $p < 0.05$, '++' $p < 0.01$ and '+++' $p < 0.001$.

Table 5: **Test accuracy** results for the **All Spans task** by evaluation regime.

| Model | New Participant | New Text | New Text & Participant |
|---|---|---|---|
| Chance / Majority Baseline | $33.2 \pm 0.5_{+++}$ | $32.7 \pm 0.6_{+++}$ | $33.3 \pm 0.9_{+++}$ |
| Question Similarity to RT-Weighted Passage | $33.3 \pm 0.4_{+++}$ | $33.4 \pm 0.4_{+++}$ | $34.9 \pm 1.4_{+++}$ |
| RT Similarity to Question-Word Similarities | $34.3 \pm 0.6_{+++}$ | $34.1 \pm 0.6^{*}_{++}$ | $34.2 \pm 1.2_{+++}$ |
| Haller RNN (Haller et al., 2022) | $43.3 \pm 1.1^{***}_{+++}$ | $40.4 \pm 0.5^{***}_{+++}$ | $40.8 \pm 1.4^{***}_{++}$ |
| RoBERTEye-Fixations (Shubi et al., 2024) | $\mathbf{49.9 \pm 0.7}^{***}$ | $\mathbf{49.0 \pm 0.8}^{***}$ | $\mathbf{47.8 \pm 1.5}^{***}$ |
| DalEye-LLaVA | $33.2 \pm 0.3_{+++}$ | $33.6 \pm 0.2_{+++}$ | $34.6 \pm 1.1_{+++}$ |
| DalEye-Llama | $43.1 \pm 1.6^{***}_{+++}$ | $35.4 \pm 0.4^{***}_{+++}$ | $39.3 \pm 1.0^{**}_{+++}$ |

Table 6: **Test accuracy** results for the **Different Spans task** variation by evaluation regime.

| Model | New Participant Different Spans | New Text Different Spans | New Text & Participant Different Spans |
|---|---|---|---|
| Chance / Majority Baseline | $55.7 \pm 0.5_{+++}$ | $55.2 \pm 0.7_{+++}$ | $53.5 \pm 1.5_{+++}$ |
| Question Similarity to RT-Weighted Passage | $54.9 \pm 3.7_{+++}$ | $55.1 \pm 3.7_{+++}$ | $57.8 \pm 3.6_{+++}$ |
| RT Similarity to Question-Word Similarities | $54.8 \pm 1.4_{+++}$ | $54.8 \pm 1.4_{+++}$ | $55.1 \pm 1.7_{+++}$ |
| Haller RNN (Haller et al., 2022) | $66.2 \pm 0.9^{***}_{+++}$ | $64.9 \pm 0.6^{***}_{+++}$ | $66.4 \pm 1.2^{***}$ |
| RoBERTEye-Fixations (Shubi et al., 2024) | $\mathbf{70.7 \pm 0.7}^{***}$ | $\mathbf{71.2 \pm 0.9}^{***}$ | $\mathbf{69.1 \pm 1.1}^{***}$ |
| DalEye-LLaVA | $55.7 \pm 0.5_{+++}$ | $58.1 \pm 1.2^{***}_{+++}$ | $58.7 \pm 1.8^{*}_{+++}$ |
| DalEye-Llama | $65.7 \pm 1.4^{***}_{+++}$ | $57.4 \pm 1.0^{*}_{+++}$ | $59.9 \pm 0.9^{**}_{+++}$ |

Table 7: **Test accuracy** results for the **Same Spans task** variation by evaluation regime.

| Model | New Participant Same Span | New Text Same Span | New Text & Participant Same Span |
|---|---|---|---|
| Chance / Majority Baseline | $50.2 \pm 0.6_{+++}$ | $49.4 \pm 0.5_{+++}$ | $51.4 \pm 1.8$ |
| Question Similarity to RT-Weighted Passage | $49.8 \pm 0.8_{+++}$ | $50.3 \pm 0.5_{+++}$ | $49.7 \pm 2.3_{++}$ |
| RT Similarity to Question-Word Similarities | $51.0 \pm 0.5_{+++}$ | $51.0 \pm 0.7_{+++}$ | $53.1 \pm 1.4$ |
| Haller RNN (Haller et al., 2022) | $52.6 \pm 1.0^{*}_{+++}$ | $51.7 \pm 0.5^{*}_{+++}$ | $50.3 \pm 2.0_{+}$ |
| RoBERTEye-Fixations (Shubi et al., 2024) | $\mathbf{58.2} \pm \mathbf{0.7}^{***}$ | $\mathbf{56.5} \pm \mathbf{0.8}^{***}$ | $\mathbf{56.7} \pm \mathbf{1.6}$ |
| DalEye-LLaVA | $49.0 \pm 0.4_{+++}$ | $49.6 \pm 0.4_{+++}$ | $51.5 \pm 1.6_{+}$ |
| DalEye-Llama | $54.2 \pm 0.7^{***}_{+++}$ | $50.4 \pm 0.3_{+++}$ | $55.0 \pm 1.5$ |

Table 8: **Validation accuracy** results for **each of the tasks**.

| Model | All Spans | Diff Span | Same Span |
|---|---|---|---|
| Chance / Majority Baseline | $33.1 \pm 0.3_{+++}$ | $55.5 \pm 0.4_{+++}$ | $50.1 \pm 0.3_{+++}$ |
| Question Similarity to RT-Weighted Passage | $33.0 \pm 0.4_{+++}$ | $54.6 \pm 3.8_{+++}$ | $50.0 \pm 0.5_{+++}$ |
| RT Similarity to Question-Word Similarities | $34.1 \pm 0.3_{+++}$ | $54.3 \pm 1.3_{+++}$ | $51.1 \pm 0.5_{+++}$ |
| Haller RNN (Haller et al., 2022) | $44.2 \pm 0.3^{***}_{+++}$ | $66.4 \pm 0.4^{***}_{+++}$ | $53.3 \pm 0.4^{***}_{+++}$ |
| RoBERTEye-Fixations (Shubi et al., 2024) | $\mathbf{50.9} \pm \mathbf{0.3}^{***}$ | $\mathbf{71.5} \pm \mathbf{0.4}^{***}$ | $\mathbf{58.5} \pm \mathbf{0.5}^{***}$ |
| DalEye-LLaVA | $34.8 \pm 0.3^{**}_{+++}$ | $56.8 \pm 0.7^{*}_{+++}$ | $50.8 \pm 0.4_{+++}$ |
| DalEye-Llama | $38.4 \pm 0.4^{***}_{+++}$ | $60.5 \pm 0.6^{***}_{+++}$ | $51.7 \pm 0.4^{*}_{+++}$ |

Table 9: **Validation accuracy** results for the **All Spans task** by evaluation regime.

| Model | New Participant | New Text | New Text & Participant |
|---|---|---|---|
| Chance / Majority Baseline | $33.0 \pm 0.6_{+++}$ | $33.3 \pm 0.5_{+++}$ | $31.7 \pm 1.2_{+++}$ |
| Question Similarity to RT-Weighted Passage | $32.6 \pm 0.3_{+++}$ | $33.2 \pm 0.7_{+++}$ | $34.7 \pm 1.6_{+++}$ |
| RT Similarity to Question-Word Similarities | $34.2 \pm 0.6_{+++}$ | $34.1 \pm 0.6_{+++}$ | $34.3 \pm 1.3_{+++}$ |
| Haller RNN (Haller et al., 2022) | $45.7 \pm 0.8^{***}_{+++}$ | $42.9 \pm 0.3^{***}_{+++}$ | $42.4 \pm 1.1^{***}_{+++}$ |
| RoBERTEye-Fixations (Shubi et al., 2024) | $\mathbf{51.3} \pm \mathbf{0.7}^{***}$ | $\mathbf{50.6} \pm \mathbf{0.8}^{***}$ | $\mathbf{50.7} \pm \mathbf{1.5}^{***}$ |
| DalEye-LLaVA | $34.8 \pm 0.6^{*}_{+++}$ | $34.3 \pm 0.3_{+++}$ | $38.5 \pm 1.5^{**}_{+++}$ |
| DalEye-Llama | $40.8 \pm 0.7^{***}_{+++}$ | $35.8 \pm 0.4^{**}_{+++}$ | $39.1 \pm 1.1^{***}_{+++}$ |

Table 10: **Validation accuracy** results for the **Different Spans task** variation by evaluation regime.

| Model | New Participant Different Spans | New Text Different Spans | New Text & Participant Different Spans |
|---|---|---|---|
| Chance / Majority Baseline | $55.4 \pm 0.6_{+++}$ | $55.8 \pm 0.6_{+++}$ | $53.2 \pm 0.9_{+++}$ |
| Question Similarity to RT-Weighted Passage | $54.4 \pm 3.9_{+++}$ | $54.6 \pm 3.8_{+++}$ | $57.3 \pm 3.8_{+++}$ |
| RT Similarity to Question-Word Similarities | $54.4 \pm 1.2_{+++}$ | $54.3 \pm 1.4_{+++}$ | $53.8 \pm 2.3_{+++}$ |
| Haller RNN (Haller et al., 2022) | $67.8 \pm 1.0^{***}_{+++}$ | $65.1 \pm 0.8^{***}_{+++}$ | $65.3 \pm 0.7^{***}_{+}$ |
| RoBERTEye-Fixations (Shubi et al., 2024) | $\mathbf{71.6 \pm 0.7}^{***}$ | $\mathbf{71.6 \pm 0.6}^{***}$ | $\mathbf{70.4 \pm 1.9}^{***}$ |
| DalEye-LLaVA | $56.4 \pm 0.8_{+++}$ | $56.6 \pm 1.1_{+++}$ | $60.8 \pm 1.9^{***}_{+++}$ |
| DalEye-Llama | $62.6 \pm 0.9^{***}_{+++}$ | $58.5 \pm 1.1^{*}_{+++}$ | $61.2 \pm 0.9^{***}_{+++}$ |

Table 11: **Validation accuracy** results for the **Same Spans task** variation by evaluation regime.

| Model | New Participant Same Span | New Text Same Span | New Text & Participant Same Span |
|---|---|---|---|
| Chance / Majority Baseline | $50.0 \pm 0.6_{+++}$ | $50.1 \pm 0.4_{+++}$ | $50.7 \pm 2.0_{++}$ |
| Question Similarity to RT-Weighted Passage | $50.6 \pm 0.8_{+++}$ | $49.3 \pm 0.6_{+++}$ | $49.9 \pm 2.2_{++}$ |
| RT Similarity to Question-Word Similarities | $51.0 \pm 0.6_{+++}$ | $50.9 \pm 0.6_{+++}$ | $53.0 \pm 1.5_{+}$ |
| Haller RNN (Haller et al., 2022) | $53.4 \pm 0.7^{**}_{+++}$ | $53.2 \pm 0.4^{**}_{+++}$ | $52.2 \pm 1.7_{+}$ |
| RoBERTEye-Fixations (Shubi et al., 2024) | $\mathbf{59.7 \pm 0.5}^{***}$ | $\mathbf{57.3 \pm 0.9}^{***}$ | $\mathbf{58.6 \pm 1.9}^{**}$ |
| DalEye-LLaVA | $50.7 \pm 0.9_{+++}$ | $51.0 \pm 0.3_{+++}$ | $49.8 \pm 1.5_{+++}$ |
| DalEye-Llama | $52.5 \pm 0.6^{*}_{+++}$ | $50.9 \pm 0.4_{+++}$ | $53.2 \pm 1.2_{+}$ |

### G.1.2 Evaluation of Generative Models on Goal Selection

For the generative models, goal selection is performed by computing the model-assigned log-likelihood of each candidate question given the input and selecting the question with the highest log-likelihood. As GPT-4o-mini and Gemini 3 do not allow extracting input token logits, we do not evaluate them on the question selection task.

Table 12: *Goal selection* test accuracy aggregated over 10 cross-validation splits, with 95% confidence intervals. We test for differences between models using a linear mixed effects model with random intercepts and slopes for participants and paragraphs: $is\_correct \sim model + (model \mid participant) + (model \mid paragraph)$. Model accuracy is compared to the chance performance of the discriminative models. Significant gains over this baseline are marked with '**' $p < 0.01$, and '***' $p < 0.001$. The best performing model in each evaluation regime is marked in bold.

| Model Type | Model | All | Different Spans | Same Span |
|---|---|---|---|---|
| Discriminative | Chance / Majority Baseline | $33.0 \pm 0.4$ | $55.3 \pm 0.4$ | $49.9 \pm 0.4$ |
| Generative | DalEye-LLaVA | $33.5 \pm 0.3$ | $57.0 \pm 0.7^{**}$ | $49.4 \pm 0.4$ |
| | DalEye-Llama | $\mathbf{39.2 \pm 0.9}^{***}$ | $\mathbf{61.4 \pm 0.9}^{***}$ | $\mathbf{52.5 \pm 0.4}^{***}$ |

The results are presented in Table 12. DalEye-Llama outperforms DalEye-LLaVA, where the latter is at chance level except for the *Different Spans* condition. The advantage of the discriminative models over the generative models is not surprising, as differently from the discriminative models, the generative models are not explicitly trained on the question selection task.

### G.1.3 RoBERTEye-Fixations Feature Ablation

To quantify the contribution of different eye-movement and textual features to the best performing discriminative model, RoBERTEye-Fixations, we conducted a feature ablation study. Here, we repeat the main experiment using the same training procedure and hyperparameter search space, ablating each of three feature groups: *fixation-level* eye movement features (e.g. current fixation duration, next saccade direction), *word-level* eye movement features (reading measures aggregated for each word, e.g. sum of all fixation durations on the word), and *linguistic* features of the words (e.g. word frequency, surprisal). This allows us to evaluate the contributions of two different eye movement representations (fixation-level and word-level) and the importance of enabling interactions of eye movement features with linguistic word characteristics. We further divide the fixation-level features into two groups: fixations and saccades, and evaluate the contribution of each group. Furthermore, we also ablate the total paragraph reading time feature. To complement this analysis, we also evaluate two additional representations: one with only linguistic features and another that includes linguistic features and total paragraph reading time feature, a measure that doesn't require an eyetracker. Table 3 presents the features of each feature group.

The feature ablation analysis results are reported in Table 13. We find that removing any of the feature groups reduces performance numerically across all evaluation settings except for ablating the text features or the saccade features in the Same Span regime. In the *Different Spans* evaluation regime, ablating any feature group, except for the word-level eye movement features, results in a statistically significant drop in performance. Notably, the drop in performance when removing either total paragraph reading time or text features indicates that the model exploits information beyond eye movement features to differentiate between questions tied to different critical spans. By contrast, in the *Same Span* evaluation regime, as candidate questions refer to the same critical span, finer-grained information than global reading times or text properties, such as fixation durations, is necessary. The largest and only statistically significant drop in performance is observed when both fixation-level feature groups are ablated jointly, with most of the effect driven by the non-saccade fixation features. Overall, the relatively small effect sizes across ablations and the similar performance of the ablations to the linguistic-features-only and linguistic features plus paragraph reading time representations suggest redundancy among feature groups, and that a substantial portion of the signal is already possibly captured by the fixation sequence via the alignment of fixations to word embeddings.

Table 13: **Feature Ablation Results** for **RoBERTEye-Fixations**. The best performing model in each evaluation regime is marked in bold. Significant drops compared to the best model are marked with '+' ($p < 0.05$) and '++' ($p < 0.01$) in subscript.

|  | All | Different Spans | Same Span |
|---|---|---|---|
| RoBERTEye-Fixations | $\mathbf{49.2 \pm 0.4}$ | $\mathbf{71.0 \pm 0.4}$ | $57.1 \pm 0.6$ |
| – Fixation-Level Features | $48.2 \pm 0.5_{+}$ | $70.0 \pm 0.5_{+}$ | $55.9 \pm 0.8_{+}$ |
| – Fixation Features | $48.4 \pm 0.5$ | $70.1 \pm 0.6_{+}$ | $56.4 \pm 0.6$ |
| – Saccade Features | $48.6 \pm 0.6$ | $69.8 \pm 0.5_{++}$ | $\mathbf{57.2 \pm 0.7}$ |
| – Word-Level Features | $48.7 \pm 0.6$ | $70.3 \pm 0.4$ | $56.7 \pm 0.7$ |
| – Linguistic Text Features | $48.6 \pm 0.6$ | $69.8 \pm 0.5_{++}$ | $\mathbf{57.2 \pm 0.7}$ |
| – Paragraph Reading Time | $48.4 \pm 0.5$ | $69.5 \pm 0.4_{++}$ | $56.9 \pm 0.7$ |
| Only Linguistic Text Features | $48.6 \pm 0.6$ | $69.9 \pm 0.4_{+}$ | $56.6 \pm 0.7$ |
| Only Linguistic Text Features + Paragraph Reading Time | $48.2 \pm 0.5_{+}$ | $69.7 \pm 0.5_{++}$ | $56.2 \pm 0.7$ |

## G.2 GOAL RECONSTRUCTION

### G.2.1 NUMERICAL RESULTS WITH STATISTICAL TESTS

Tables 14 and 15 present the goal reconstruction evaluations of Figure 5 in numerical format. Presented are question reconstruction evaluations of DalEye-Llama, DalEye-GPT and Gemini (zero-shot and few-shot) for (1) the identity of the generated question word, (2) UIUC semantic category of the question, (3) BLEU score, (4) BERTScore, (5) downstream QA accuracy based on the answer selection of a multiple-choice QA model. We additionally provide evaluations for DalEye-LLaVA. The models are benchmarked against six baselines that *do not include eye movements*. Two baselines are human-composed questions for a different and the same critical span. Four additional baselines are LLM-generated questions: an arbitrary question about the text generated with Gemini 3, and questions from text-only variants of DalEye-Llama, DalEye-GPT and DalEye-LLaVA, finetuned on the question decoding task using only the text.

Table 14: Goal reconstruction evaluations on the **Question Word accuracy**, **Question Category accuracy** and **BLEU** measures. Presented are intercept coefficients with 95% confidence intervals from a linear mixed effects model with random intercepts for participants and paragraphs: $measure \sim 1 + (1 \mid participant) + (1 \mid paragraph)$ for eye movement models and the $measure \sim 1 + (1 \mid paragraph)$ for the baselines. The highest score in each combination of evaluation measure and evaluation regime is marked in bold. Model comparisons were conducted using $measure \sim model + (1 \mid participant) + (1 \mid paragraph)$. Significant drops compared to the best model are marked in superscript with '*' $p < 0.05$, '**' $p < 0.01$, and '***' $p < 0.001$. Significant improvements over corresponding baselines are marked in subscript with '+' $p < 0.05$, '++' $p < 0.01$, and '+++' $p < 0.001$.

| | New Participant | New Item | New Participant & Item |
|---|---|---|---|
| **Question Word accuracy** | | | |
| Incorrect Human (different critical span) | $0.416 \pm 0.022^{***}$ | $0.407 \pm 0.022^{***}$ | $0.403 \pm 0.024^{***}$ |
| Incorrect Human (same critical span) | $0.370 \pm 0.019^{***}$ | $0.370 \pm 0.019^{***}$ | $0.401 \pm 0.021^{***}$ |
| Text-only Llama 3.1 | $0.591 \pm 0.014^{***}$ | $0.540 \pm 0.018$ | $0.540 \pm 0.020$ |
| DalEye-Llama | $0.674 \pm 0.013^{***}_{+++}$ | $0.545 \pm 0.017$ | $0.548 \pm 0.023$ |
| Text-only GPT-4o-mini | $0.561 \pm 0.014^{***}$ | $0.450 \pm 0.014^{***}$ | $0.479 \pm 0.020^{***}$ |
| DalEye-GPT | $0.617 \pm 0.014^{***}_{+++}$ | $0.495 \pm 0.015^{***}_{+++}$ | $0.496 \pm 0.023^{***}$ |
| Arbitrary Gemini 3 | $0.318 \pm 0.013^{***}$ | $0.327 \pm 0.013^{***}$ | $0.328 \pm 0.019^{***}$ |
| Gemini zero-shot | $0.382 \pm 0.012^{***}_{+++}$ | $0.378 \pm 0.013^{***}_{+++}$ | $0.393 \pm 0.022^{***}_{+++}$ |
| Gemini few-shot | $\mathbf{0.713 \pm 0.012}_{+++}$ | $0.395 \pm 0.013^{***}_{+++}$ | $0.410 \pm 0.022^{***}_{+++}$ |
| Text-only LLaVA OneVision | $0.538 \pm 0.017^{***}$ | $0.499 \pm 0.019^{***}$ | $0.498 \pm 0.021^{***}$ |
| DalEye-LLaVA | $0.558 \pm 0.018^{***}_{+++}$ | $\mathbf{0.548 \pm 0.019}_{+++}$ | $\mathbf{0.548 \pm 0.024}_{+++}$ |
| **Question Category accuracy** | | | |
| Incorrect Human (different critical span) | $0.410 \pm 0.021^{***}$ | $0.410 \pm 0.021^{***}$ | $0.423 \pm 0.024^{*}$ |
| Incorrect Human (same critical span) | $0.349 \pm 0.019^{***}$ | $0.349 \pm 0.019^{***}$ | $0.357 \pm 0.021^{***}$ |
| Text-only Llama 3.1 | $0.535 \pm 0.014^{***}$ | $0.454 \pm 0.017^{***}$ | $0.453 \pm 0.021$ |
| DalEye-Llama | $\mathbf{0.558 \pm 0.014}_{+++}$ | $0.450 \pm 0.017^{***}$ | $0.428 \pm 0.023^{***}$ |
| Text-only GPT-4o-mini | $0.458 \pm 0.013^{***}$ | $0.430 \pm 0.014^{***}$ | $0.429 \pm 0.020^{**}$ |
| DalEye-GPT | $0.550 \pm 0.013_{+++}$ | $\mathbf{0.488 \pm 0.015}_{+++}$ | $0.454 \pm 0.022_{++}$ |
| Arbitrary Gemini 3 | $0.345 \pm 0.013^{***}$ | $0.367 \pm 0.014^{***}$ | $0.355 \pm 0.019^{***}$ |
| Gemini zero-shot | $0.327 \pm 0.012^{***}$ | $0.335 \pm 0.012^{***}$ | $0.323 \pm 0.021^{***}$ |
| Gemini few-shot | $0.526 \pm 0.014^{***}_{+++}$ | $0.418 \pm 0.013^{***}_{+++}$ | $0.439 \pm 0.022^{**}_{++}$ |
| Text-only LLaVA OneVision | $0.408 \pm 0.013^{***}$ | $0.424 \pm 0.018^{***}$ | $0.412 \pm 0.021^{***}$ |
| DalEye-LLaVA | $0.445 \pm 0.015^{***}_{+++}$ | $0.466 \pm 0.019^{***}_{+++}$ | $\mathbf{0.460 \pm 0.024}_{+++}$ |
| **BLEU** | | | |
| Incorrect Human (different critical span) | $0.219 \pm 0.006^{***}$ | $0.219 \pm 0.006^{***}$ | $0.219 \pm 0.007^{***}$ |
| Incorrect Human (same critical span) | $0.298 \pm 0.009^{***}$ | $\mathbf{0.298 \pm 0.009}$ | $\mathbf{0.300 \pm 0.009}$ |
| Text-only Llama 3.1 | $0.454 \pm 0.006^{***}$ | $0.236 \pm 0.004^{***}$ | $0.238 \pm 0.005^{***}$ |
| DalEye-Llama | $0.556 \pm 0.008^{***}_{+++}$ | $0.253 \pm 0.005^{***}_{+++}$ | $0.244 \pm 0.006^{***}$ |
| Text-only GPT-4o-mini | $0.452 \pm 0.006^{***}$ | $0.240 \pm 0.004^{***}$ | $0.237 \pm 0.006^{***}$ |
| DalEye-GPT | $0.514 \pm 0.008^{***}_{+++}$ | $0.257 \pm 0.004^{***}_{+++}$ | $0.256 \pm 0.007^{***}_{+++}$ |
| Arbitrary Gemini 3 | $0.191 \pm 0.004^{***}$ | $0.194 \pm 0.005^{***}$ | $0.184 \pm 0.006^{***}$ |
| Gemini zero-shot | $0.246 \pm 0.005^{***}_{+++}$ | $0.247 \pm 0.005^{***}_{+++}$ | $0.242 \pm 0.009^{***}_{+++}$ |
| Gemini few-shot | $\mathbf{0.644 \pm 0.009}_{+++}$ | $0.272 \pm 0.005^{***}_{+++}$ | $0.268 \pm 0.008^{***}_{+++}$ |
| Text-only LLaVA OneVision | $0.304 \pm 0.005^{***}$ | $0.210 \pm 0.004^{***}$ | $0.207 \pm 0.005^{***}$ |
| DalEye-LLaVA | $0.265 \pm 0.005^{***}$ | $0.231 \pm 0.006^{***}_{+++}$ | $0.231 \pm 0.007^{***}_{+++}$ |

Table 15: Goal reconstruction evaluations on the **BERTScore** and **QA accuracy** measures. Presented are intercept coefficients with 95% confidence intervals from a linear mixed effects model with random intercepts for participants and paragraphs: $measure \sim 1 + (1 \mid participant) + (1 \mid paragraph)$ for eye movement models and the $measure \sim 1 + (1 \mid paragraph)$ for the baselines. The highest score in each combination of evaluation measure and evaluation regime is marked in bold. Model comparisons were conducted using $measure \sim model + (1 \mid participant) + (1 \mid paragraph)$. Significant drops compared to the best model are marked in superscript with '*' $p < 0.05$, '**' $p < 0.01$, and '***' $p < 0.001$. Significant improvements over corresponding baselines are marked in subscript with '+' $p < 0.05$, '++' $p < 0.01$, and '+++' $p < 0.001$.

| | New Participant | New Item | New Participant & Item |
|---|---|---|---|
| **BERTScore** | | | |
| Incorrect Human (different critical span) | $0.604 \pm 0.004^{***}$ | $0.603 \pm 0.004^{***}$ | $0.602 \pm 0.004^{***}$ |
| Incorrect Human (same critical span) | $0.651 \pm 0.004^{***}$ | $\mathbf{0.651 \pm 0.004}$ | $\mathbf{0.651 \pm 0.004}$ |
| Text-only Llama 3.1 | $0.723 \pm 0.003^{***}$ | $0.617 \pm 0.003^{***}$ | $0.618 \pm 0.003^{***}$ |
| DalEye-Llama | $0.776 \pm 0.004^{***}_{+++}$ | $0.631 \pm 0.003^{***}_{+++}$ | $0.626 \pm 0.004^{***}_{+++}$ |
| Text-only GPT-4o-mini | $0.722 \pm 0.003^{***}$ | $0.619 \pm 0.003^{***}$ | $0.618 \pm 0.004^{***}$ |
| DalEye-GPT | $0.755 \pm 0.004^{***}_{+++}$ | $0.630 \pm 0.003^{***}_{+++}$ | $0.627 \pm 0.004^{***}_{++}$ |
| Arbitrary Gemini 3 | $0.611 \pm 0.003^{***}$ | $0.612 \pm 0.003^{***}$ | $0.605 \pm 0.003^{***}$ |
| Gemini zero-shot | $0.628 \pm 0.003^{***}_{+++}$ | $0.629 \pm 0.003^{***}_{+++}$ | $0.627 \pm 0.005^{***}_{+++}$ |
| Gemini few-shot | $\mathbf{0.818 \pm 0.005}_{+++}$ | $0.642 \pm 0.003^{***}_{+++}$ | $0.644 \pm 0.005^{***}_{+++}$ |
| Text-only LLaVA OneVision | $0.647 \pm 0.003^{***}$ | $0.595 \pm 0.003^{***}$ | $0.591 \pm 0.003^{***}$ |
| DalEye-LLaVA | $0.638 \pm 0.003^{***}$ | $0.618 \pm 0.003^{***}_{+++}$ | $0.617 \pm 0.004^{***}_{+++}$ |
| **QA accuracy** | | | |
| Incorrect Human (different critical span) | $0.495 \pm 0.018^{***}$ | $0.490 \pm 0.017^{***}$ | $0.490 \pm 0.020^{***}$ |
| Incorrect Human (same critical span) | $0.677 \pm 0.021^{***}$ | $0.677 \pm 0.021$ | $0.686 \pm 0.022$ |
| Text-only Llama 3.1 | $0.684 \pm 0.012^{***}$ | $0.609 \pm 0.014^{***}$ | $0.616 \pm 0.020^{***}$ |
| DalEye-Llama | $0.763 \pm 0.011^{***}_{+++}$ | $0.648 \pm 0.014^{***}_{+++}$ | $0.644 \pm 0.023^{*}_{+}$ |
| Text-only GPT-4o-mini | $0.679 \pm 0.012^{***}$ | $0.619 \pm 0.013^{***}$ | $0.608 \pm 0.020^{***}$ |
| DalEye-GPT | $0.738 \pm 0.011^{***}_{+++}$ | $0.658 \pm 0.013^{***}_{+++}$ | $0.644 \pm 0.023^{***}$ |
| Arbitrary Gemini 3 | $0.635 \pm 0.015^{***}$ | $0.636 \pm 0.015^{***}$ | $0.614 \pm 0.020^{***}$ |
| Gemini zero-shot | $0.674 \pm 0.013^{***}_{+++}$ | $0.664 \pm 0.013^{***}_{+++}$ | $\mathbf{0.690 \pm 0.022}_{+++}$ |
| Gemini few-shot | $\mathbf{0.804 \pm 0.010}_{+++}$ | $\mathbf{0.683 \pm 0.012}_{+++}$ | $0.671 \pm 0.023^{*}_{+++}$ |
| Text-only LLaVA OneVision | $0.632 \pm 0.012^{***}$ | $0.636 \pm 0.014^{***}$ | $0.612 \pm 0.019^{***}$ |
| DalEye-LLaVA | $0.623 \pm 0.013^{***}$ | $0.610 \pm 0.016^{***}$ | $0.579 \pm 0.024^{***}$ |

### G.2.2 EXAMPLE GENERATIONS

Example generations for DalEye-Llama, DalEye-GPT, Gemini zero-shot, and Gemini few-shot are presented in Table 16. The complete set of model outputs for all three evaluation regimes, including the baselines, together with the corresponding annotations, is released in the project's GitHub repository under `generated_questions.csv`.

Table 16: An example showing a paragraph, the corresponding ground truth question, and, for DalEye-Llama, Gemini (zero-shot and few-shot), and DalEye-GPT, four generated questions from the New Text Regime and four from the New Participant Regime. Each generated question reflects a different outcome from the QA model, as indicated by the answer choice (A–D) selected by the model. "(no example)" indicates that the model did not generate a question for the corresponding answer choice. The possible answers, structured according to the STARC annotation guidelines (Berzak et al., 2020), to the ground truth question were: A) (correct answer) A large number of cultural destinations. B) (miscomprehension of the critical span) Beaches constantly full of locals and tourists. C) (incorrect, related to a different span) 3,000 luxury hotels. D) (no support in the passage) Highly-regarded restaurants.

**Paragraph:** *Lemnos has wild beaches, where you can swim and sunbathe almost alone, a small nightlife scene and many cultural sites. Lemnos is the eighth largest island in Greece so it will have to pay the first round of tax increases in autumn 2015. But Lemnos is far less wealthy than many smaller islands. It has just over 3,000 beds for visitors – Rhodes, for example, has tens of thousands of beds. "We have been suffering economically in recent years and now we will suffer more," said Lemnos Mayor, Dimitris Marinakis.*

**True Question:** *What is among the primary attractions of Lemnos?*

| Model | New Participant Generated Questions | New Text Generated Questions |
|---|---|---|
| **DalEye-Llama** | (A) What is among the primary attractions of Lemnos? | (A) What is one thing to note about the eponymous village? |
| | (B) What will happen in spring 2013? | (B) Who is quoted as saying that hotels benefit from the increase in the number of tourists in the area? |
| | (C) Why does Lemnos have to pay the first round of tax increases? | (C) What does the quote convey about the current economic situation in Greece? |
| | (D) What is one reason why the wealthy tend to spend more time on the islands? | (D) What is true of the other Greek islands? |
| **Gemini Zero-Shot** | (A) What can you do on the wild beaches of Lemnos? | (A) What does Lemnos have? |
| | (B) Where can you swim and sunbathe almost alone? | (B) What kind of nightlife does Lemnos have? |
| | (C) Why is Lemnos in line for the first round of tax increases? | (C) When will Lemnos have to pay the first round of tax increases? |
| | (D) (no example) | (D) Are the beaches on Lemnos crowded in the summer? |
| **Gemini Few-Shot** | (A) What is among the primary attractions of Lemnos? | (A) What does Lemnos have dozens of? |
| | (B) (no example) | (B) (no example) |
| | (C) (no example) | (C) Why is Lemnos in line for the first round of tax increases? |
| | (D) Why does Lemnos have to pay the first round of tax increases? | (D) Why will Lemnos have to pay the first round of tax increases? |
| **DalEye-GPT** | (A) What is among the primary attractions of Lemnos? | (A) What does Lemnos have dozens of? |
| | (B) (no example) | (B) What is true of Rhodes? |
| | (C) Why does Lemnos have to pay the first round of tax increases? | (C) What does Dimitris Marinakis say about the current economic situation on Lemnos? |
| | (D) Why does Lemnos have to pay the first round of tax increases? | (D) What is one thing Lemnos' councilors say they will do? |

### G.2.3 DALEYE-LLAMA: PROMPT AND EYE MOVEMENT REPRESENTATION VARIATIONS

We experiment with three input representations that capture different granularities of eye movement information: (i) Fixation—level, (ii) Word-level, and (iii) a combination of both fixation- and word-level information, as detailed below:

1. **Fixations: Fixation Duration + Next saccade direction**
   - Format: `a list of fixation-level features: [fixated word index, fixated word, fixation duration in ms, direction of next saccade (backward to earlier word / within word / forward to later word)]`
   - Example: `[[4, "fox", 220, backward], ...]`

2. **Words: Total Fixation Duration + Incoming/Outgoing Backward/Forward Saccades**
   - Format: `a list of word-level features: [word index, word, total fixation duration in ms, incoming forward saccades (from earlier word), incoming backward saccades (from later word), outgoing forward saccades (to later word), outgoing backward saccades (to earlier word)]`
   - Example: `[[4, "fox", 320, 2, 1, 3, 0], ...]`

3. **Words+Fixations**
   - Format: `two lists of word-level and fixation-level features: [fixated word index, fixated word, fixation duration in ms, direction of next saccade (backward / within / forward)] [word index, word, total fixation duration in ms, incoming forward saccades, incoming backward saccades, outgoing forward saccades, outgoing backward saccades]`
   - Example: `[[4, "fox", 220, backward], ...] [[4, "fox", 320, 2, 1, 3, 0], ...]`

We further experiment with an alternative prompt that de-emphasizes the experimental setup and places more focus on the generation task itself:

```
"""
Task Description:
Your task is to generate the exact original question a
reader had in mind before reading a given paragraph.
The input data is (a/two) time series composed of
(word-level/fixation-level/word-level and fixation-level)
features.

Input Format:
You will receive the paragraph and eye movement data formatted as
{FORMAT}

Expected Output:
Generate the exact original question provided to the reader,
accurately inferred from the fixation patterns.

Instructions:
Your output must precisely match the original question presented
to the reader.
Produce only the exact original question as your output.

{PARAGRAPH}
{SCANPATH}
"""
```

In Table 17 we present the question selection results for DalEye-Llama across the two proposed task prompts and the three input representations. Across all three task variations, Main task prompt with fixation-level input yields the best accuracy. Both Alternative task prompt and the word-level/combined inputs underperform this setting, where the choice of representation matters more than the task prompt. In Table 18 we present goal reconstruction results that similarly suggest that the alternative prompt and eye movements representations for DalEye-Llama weaken its performance in most cases.

Table 17: **Test accuracy** results for **each of the tasks** and for **each of the input variations**.

| Task Prompt | Representation | All Spans | Diff Span | Same Span |
|---|---|---|---|---|
| Chance / Majority Baseline | None | $33.0 \pm 0.4_{+++}$ | $55.3 \pm 0.4_{+++}$ | $49.9 \pm 0.4_{+++}$ |
| Main task prompt | Fixation-level | $\mathbf{39.2 \pm 0.9}^{***}$ | $\mathbf{61.4 \pm 0.9}^{***}$ | $\mathbf{52.5 \pm 0.4}^{***}$ |
| | Word-level | $33.9 \pm 0.5_{+++}$ | $55.5 \pm 0.6_{+++}$ | $50.4 \pm 0.3_{++}$ |
| | Both fixation- and word-level | $34.7 \pm 0.6^{***}_{+++}$ | $56.2 \pm 0.8_{+++}$ | $50.5 \pm 0.4_{++}$ |
| Alternative task prompt | Fixation-level | $37.2 \pm 0.6^{***}_{+++}$ | $60.3 \pm 0.6^{***}$ | $51.0 \pm 0.3_{+}$ |
| | Word-level | $33.4 \pm 0.2_{+++}$ | $55.0 \pm 0.6_{+++}$ | $50.2 \pm 0.2_{+++}$ |
| | Both fixation- and word-level | $34.4 \pm 0.7^{**}_{+++}$ | $55.5 \pm 1.0_{+++}$ | $50.5 \pm 0.5_{++}$ |

Table 18: Test evaluations of the DalEye-Llama model with the fixation-level, the word-level, and combined word- and fixation-level input representations, on two task prompt variants.

| | | New Participant | New Item | New Participant & Item |
|---|---|---|---|---|
| **Question Word accuracy** | | | | |
| Main task prompt | Fixation-level | **0.674 ± 0.013** | **0.545 ± 0.017** | **0.548 ± 0.023** |
| | Word-level | 0.185 ± 0.019 | 0.128 ± 0.014 | 0.125 ± 0.016 |
| | Both fixation- and word-level | 0.191 ± 0.020 | 0.130 ± 0.014 | 0.134 ± 0.016 |
| Alternative task prompt | Fixation-level | 0.635 ± 0.014 | 0.534 ± 0.017 | 0.547 ± 0.023 |
| | Word-level | 0.182 ± 0.018 | 0.129 ± 0.014 | 0.133 ± 0.016 |
| | Both fixation- and word-level | 0.134 ± 0.017 | 0.091 ± 0.011 | 0.099 ± 0.015 |
| **Question Category accuracy** | | | | |
| Main task prompt | Fixation-level | **0.558 ± 0.014** | 0.450 ± 0.017 | 0.428 ± 0.023 |
| | Word-level | 0.448 ± 0.015 | 0.432 ± 0.015 | 0.438 ± 0.022 |
| | Both fixation- and word-level | 0.286 ± 0.015 | 0.257 ± 0.015 | 0.259 ± 0.020 |
| Alternative task prompt | Fixation-level | 0.550 ± 0.015 | 0.484 ± 0.016 | 0.473 ± 0.024 |
| | Word-level | 0.537 ± 0.018 | **0.538 ± 0.019** | **0.529 ± 0.024** |
| | Both fixation- and word-level | 0.515 ± 0.018 | 0.507 ± 0.018 | 0.498 ± 0.024 |
| **BLEU** | | | | |
| Main task prompt | Fixation-level | **0.556 ± 0.008** | 0.253 ± 0.005 | 0.244 ± 0.006 |
| | Word-level | 0.140 ± 0.014 | 0.089 ± 0.005 | 0.085 ± 0.006 |
| | Both fixation- and word-level | 0.146 ± 0.015 | 0.088 ± 0.005 | 0.085 ± 0.006 |
| Alternative task prompt | Fixation-level | 0.493 ± 0.009 | **0.264 ± 0.004** | **0.262 ± 0.007** |
| | Word-level | 0.126 ± 0.013 | 0.077 ± 0.005 | 0.073 ± 0.006 |
| | Both fixation- and word-level | 0.096 ± 0.012 | 0.065 ± 0.004 | 0.063 ± 0.005 |
| **BERTScore** | | | | |
| Main task prompt | Fixation-level | **0.776 ± 0.004** | 0.631 ± 0.003 | 0.626 ± 0.004 |
| | Word-level | 0.429 ± 0.012 | 0.410 ± 0.007 | 0.408 ± 0.009 |
| | Both fixation- and word-level | 0.426 ± 0.013 | 0.406 ± 0.007 | 0.402 ± 0.010 |
| Alternative task prompt | Fixation-level | 0.746 ± 0.005 | **0.633 ± 0.003** | **0.632 ± 0.004** |
| | Word-level | 0.410 ± 0.013 | 0.388 ± 0.007 | 0.384 ± 0.009 |
| | Both fixation- and word-level | 0.409 ± 0.010 | 0.393 ± 0.005 | 0.390 ± 0.006 |
| **QA accuracy** | | | | |
| Main task prompt | Fixation-level | **0.763 ± 0.011** | **0.648 ± 0.014** | 0.644 ± 0.023 |
| | Word-level | 0.600 ± 0.014 | 0.569 ± 0.015 | 0.557 ± 0.024 |
| | Both fixation- and word-level | 0.619 ± 0.014 | 0.595 ± 0.015 | 0.582 ± 0.024 |
| Alternative task prompt | Fixation-level | 0.717 ± 0.012 | 0.646 ± 0.014 | **0.670 ± 0.022** |
| | Word-level | 0.577 ± 0.015 | 0.560 ± 0.015 | 0.537 ± 0.023 |
| | Both fixation- and word-level | 0.565 ± 0.016 | 0.541 ± 0.015 | 0.538 ± 0.024 |

