# OpenReview forum: "Decoding Open-Ended Information Seeking Goals from Eye Movements in Reading"
_ICLR.cc/2026/Conference — ICLR 2026 Poster_

### Official Review · Reviewer_H8io · 2025-10-31

**Soundness:** 3
**Presentation:** 3
**Contribution:** 2
**Rating:** 4
**Confidence:** 5

**Summary:**

This paper tackles the task of decoding open-ended reading goals from eye movements. Using the OneStop dataset, the authors propose two tasks-target selection and target reconstruction—and design multimodal models combining gaze and text features. Results show that gaze–text integration slightly improves performance, indicating that eye movements encode semantic goal information. This paper provides a new research paradigm for the combination of eye movement and language models, but further improvement is needed in terms of interpretability, cross-text generalization, and comparison with more powerful baselines. Overall, the paper is interesting, but its current form cannot totally support all authors' claims.

**Strengths:**

- The authors claim that they are the first study to decode open-ended, text-specific reading goals from eye movements, framed as dual tasks of selection and reconstruction.
- Integrating text and gaze features markedly improves target selection; RoBERTEye-Fixations achieves 49.3% accuracy vs. 33% baseline.
- DalEye-Llama attains 76.3% QA accuracy on unseen participants (vs. 68.1% for human distractors), validating the gaze–goal correspondence.

**Weaknesses:**

1. Lacks cognitive interpretation of gaze behavior and its link to goal decoding.
2. Sharp performance drop on unseen texts (Kappa 0.478 → 0.069) unexplained.
3. No comparison with fine-tuned multimodal LLMs (e.g., GPT-4o, LLaVA-1.5).

**Questions:**

1. What causes DalEye-Llama's performance to drop dramatically on new text? Is it due to limitations in feature transfer or model overfitting?
2. The paper mentions "inherent noise" in eye tracking data but does not discuss mitigation strategies. Have you tested noise reduction or fixation filtering to improve model stability?
3. The generation model is only compared to GPT-4o in zero-shot mode. Could results with a fine-tuned GPT-4o (using the same text and gaze inputs) clarify advantages in efficiency or robustness?

---

> ### Author Response · Authors · 2025-12-03
>
> Thank you for the thoughtful and detailed comments. We appreciate your feedback and address each point below.
>
> > Lacks cognitive interpretation of gaze behavior and its link to goal decoding.
>
> Thank you for raising this point. Following your suggestion we added a new analysis that ties model performance to cognitive aspects of gaze behavior during goal decoding (Section 7 in the revised paper).
>
> The analysis is based on the framework proposed by Shubi et al. (2025), and examines features of the textual item and the reader from the psycholinguistic literature which are pertinent to cognitive theory of information seeking (e.g. Kaakinen et al. 2022, Hahn and Keller 2023, Shubi 2023). It takes advantage of manual annotations of the Critical Span for each question to examine how these features, which are not provided to the model, affect model performance. Specifically, we use the following linear mixed effects model which accounts for the non i.i.d structure of the data:
>
> P(is_correct) ~
>     reading_time_before_CriticalSpan +
>     reading_time_in_CriticalSpan +
>     reading_time_after_CriticalSpan +
>     paragraph_position +
>     answered_correctly +
>     paragraph_length +
>     paragraph_level +
>     CriticalSpan_start_location +
>     CriticalSpan_length +
>     question_CriticalSpan_lexical_overlap +
>     (1 | participant_id) +
>     (1 | unique_paragraph_id)
>
> Where P(correct) is the probability assigned by RoBERTEye to the correct question given an input of a paragraph and eye movements.
> We then examine the sign and statistical significance of the corresponding coefficients of the features:
>
> | Feature                                   | Coefficient              |
> |:------------------------------------------|:-------------------------:|
> | Reading Time Before Critical Span         | $-0.068^{***}$           |
> | Reading Time In Critical Span             | $0.139^{***}$            |
> | Reading Time After Critical Span          | $-0.073^{***}$           |
> | Paragraph Position                        | $0.011^{**}$             |
> | Answered Correctly                        | $0.010^{*}$              |
> | Paragraph Length                          | $0.029^{***}$            |
> | Paragraph Level (original / simplified)   | $-0.007$   |
> | Critical Span Start Location              | $0.005$    |
> | Question–Critical Span Lexical Overlap    | $-0.007$   |
>
>
> In the psycholinguistic literature, studies have observed longer reading times in the Critical Span compared to outside it, which was previously linked to rational processing during information seeking (Kaakinen et al. 2002, Hahn and Keller 2023, Shubi 2023). Intriguingly, we find that the extent to which this holds affects model performance: longer reading times before and after the Critical Span hurt model performance (see the corresponding negative feature coefficients -0.068*** and -0.073***), while longer reading times within the Critical Span contribute to it (0.139***). In other words, the more goal directed and resource efficient the reader is, the easier it is to identify their goal automatically.
>
> Further in line with prior psycholinguistic findings, we find that paragraph position (due to adaptation to the task over time) and reading comprehension (task performance) both contribute to model performance. Finally, paragraph length is also a significant feature, likely as it allows better separation between question relevant and irrelevant material, as well as between critical spans of different questions. Overall, these findings confirm and expand prior observations from the cognitive literature on information using the developed model as an analytical tool.
>
> ### References
>
> Kaakinen, Johanna K., Jukka Hyönä, and Janice M. Keenan. "Perspective effects on online text processing." Discourse processes 33.2 (2002): 159-173.
>
> Hahn, Michael, and Frank Keller. "Modeling task effects in human reading with neural network-based attention." Cognition 230 (2023).
>
> Shubi, Omer, and Yevgeni Berzak. "Eye movements in information-seeking reading." Proceedings of the annual meeting of the cognitive science society. Vol. 45. No. 45. (2023).
>
> Shubi, Omer, Cfir Avraham Hadar, and Yevgeni Berzak. "Decoding reading goals from eye movements." Proceedings of the 63rd Annual Meeting of the Association for Computational Linguistics. (2025).

---

> ### Author Response · Authors · 2025-12-03
>
> > Sharp performance drop on unseen texts (Kappa 0.478 → 0.069) unexplained.
>
> > What causes DalEye-Llama's performance to drop dramatically on new text? Is it due to limitations in feature transfer or model overfitting?
>
> Thank you for this question. First, we examined the learning curves on the training and validation sets, and did not find evidence for model overfitting. We believe that this performance drop is rooted in the combination of the New Item evaluation regime difficulty and the question reconstruction task difficulty which lead to inefficient feature transfer. In the New Text regime, the model encounters completely new textual materials and questions. This stands in stark contrast to the New Participant regime, where the model has already seen both the text and the possible questions during training, a considerably easier generalization problem. Furthermore, the generation task is inherently more difficult than the discriminative question selection task, where models do generalize well to the New Text regime (see results in Table 2). Thus, we believe that the combination of the task and the evaluation regime lead the Llama model to fail to generalize in this case. We clarify this point in the revised manuscript in the question reconstruction results in Section 6.2.

---

> ### Author Response · Authors · 2025-12-03
>
> > No comparison with fine-tuned multimodal LLMs (e.g., GPT-4o, LLaVA-1.5).
>
> > The generation model is only compared to GPT-4o in zero-shot mode. Could results with a fine-tuned GPT-4o (using the same text and gaze inputs) clarify advantages in efficiency or robustness?
>
> Thank you for this suggestion. To the best of our knowledge, there is currently no off-the-shelf multimodal LLM that is generative and is specifically designed to jointly process eye-movement data and text. We address this gap by adapting a vision and language model, LLaVA-OneVision, to our modality and task. LLaVA-OneVision is a more recent variant of LLaVA-1.5 which you propose.
>
> Following your recommendation, we now also fine-tuned GPT-4o-mini using the same input representation as DalEye-Llama in the same 10-fold cross validation setup. We selected the mini version due to budget constraints, as fine-tuning GPT-4o on our task costs over 3k USD for a single experiment (the cost of training GPT-4o-mini is 300 USD). The results, presented below and in the revised paper, show that overall, GPT-4o-mini achieves comparable results to DalEye-Llama.
>
> Details about GPT-4o’s architecture, total parameter count, and the number of trainable parameters exposed for fine-tuning are not publicly available, so we are not able to comment directly on resource efficiency relative to a Llama based model. We can say with certainty that Llama has the advantage of being open and free to use.
>
> || New Participant | New Item| New Participant & Item |
> | -------------------------------------------------- | :-----------------: | :-----------------: | :----------------------: |
> | **Question word (Kappa)**| | ||
> | Incorrect human question – different critical span | 0.095 ± 0.009 | 0.098 ± 0.009 | 0.088 ± 0.026|
> | Incorrect human question – same critical span| 0.087 ± 0.014 | 0.068 ± 0.013 | 0.107 ± 0.040|
> | GPT-4o arbitrary question| 0.020 ± 0.008 | 0.025 ± 0.007 | 0.016 ± 0.019|
> | Fine-tuned GPT-4o question | 0.407 ± 0.011 | **0.117 ± 0.011** | **0.130 ± 0.027** |
> | Llama 3.1 arbitrary question | 0.346 ± 0.011 | 0.012 ± 0.005 | 0.003 ± 0.013|
> | DalEye-Llama question| **0.478 ± 0.011** | 0.069 ± 0.008 | 0.052 ± 0.022|
> | **Question category (Kappa)**| | ||
> | Incorrect human question – different critical span | 0.097 ± 0.009 | 0.107 ± 0.009 | **0.133 ± 0.028**|
> | Incorrect human question – same critical span| 0.063 ± 0.014 | 0.046 ± 0.013 | 0.104 ± 0.040|
> | GPT-4o arbitrary question| 0.005 ± 0.006 | 0.018 ± 0.006 | 0.022 ± 0.019|
> | Fine-tuned GPT-4o question | **0.318 ± 0.011** | **0.148 ± 0.011** | 0.138 ± 0.029 |
> | Llama 3.1 arbitrary question | 0.250 ± 0.011 | 0.028 ± 0.009 | 0.024 ± 0.026|
> | DalEye-Llama question| 0.299 ± 0.011 | 0.074 ± 0.010 | 0.045 ± 0.028|
> | **BLEU** | | ||
> | Incorrect human question – different critical span | 0.291 ± 0.002 | 0.291 ± 0.002 | 0.286 ± 0.007|
> | Incorrect human question – same critical span| 0.384 ± 0.004 | **0.380 ± 0.004** | **0.379 ± 0.011**|
> | GPT-4o arbitrary question| 0.212 ± 0.002 | 0.217 ± 0.002 | 0.210 ± 0.006|
> | Fine-tuned GPT-4o question | 0.572 ± 0.006 | 0.328 ± 0.003 | 0.329 ± 0.007 |
> | Llama 3.1 arbitrary question | 0.523 ± 0.005 | 0.299 ± 0.003 | 0.309 ± 0.007|
> | DalEye-Llama question| **0.602 ± 0.006** | 0.315 ± 0.003 | 0.314 ± 0.007|
> | **BERTScore**| | ||
> | Incorrect human question – different critical span | 0.604 ± 0.001 | 0.603 ± 0.001 | 0.601 ± 0.003|
> | Incorrect human question – same critical span| 0.651 ± 0.002 | **0.650 ± 0.002** | **0.654 ± 0.006**|
> | GPT-4o arbitrary question| 0.413 ± 0.005 | 0.398 ± 0.005 | 0.313 ± 0.013|
> | Fine-tuned GPT-4o question | 0.572 ± 0.006 | 0.328 ± 0.003 | 0.329 ± 0.007 |
> | Llama 3.1 arbitrary question | 0.723 ± 0.003 | 0.617 ± 0.001 | 0.617 ± 0.003|
> | DalEye-Llama question| **0.774 ± 0.003** | **0.631 ± 0.002** | **0.628 ± 0.004**|
> | **QA accuracy**| | ||
> | Incorrect human question – different critical span | 49.7 ± 0.9| 49.2 ± 0.8| 48.7 ± 2.3 |
> | Incorrect human question – same critical span| 68.1 ± 1.1| **67.7 ± 1.1**| 66.3 ± 3.0 |
> | GPT-4o arbitrary question| 67.3 ± 0.8| 66.7 ± 0.8| **66.5 ± 2.2** |
> | Fine-tuned GPT-4o question | 74.0 ± 0.7 | 65.9 ± 0.7 | 62.0 ± 2.3 |
> | Llama 3.1 arbitrary question | 68.6 ± 0.8| 60.6 ± 0.8| 62.0 ± 2.3 |
> | DalEye-Llama question| **76.3 ± 0.8**| 65.1 ± 0.8| 65.2 ± 2.2 |

---

> ### Author Response · Authors · 2025-12-03
>
> > The paper mentions "inherent noise" in eye tracking data but does not discuss mitigation strategies. Have you tested noise reduction or fixation filtering to improve model stability?
>
> Thank you for raising this point, we appreciate the opportunity to clarify it. Eye movement recordings contain two main sources of variability: (1) noise from the recording system and (2) natural behavioral variability from participants. With respect to (2), our data was recorded using a state-of-the-art EyeLink 1000 Plus system (1000 Hz sampling rate and average calibration error below 0.3 degrees, i.e. less than half a letter), so noise in the data quality sense is as low as one can currently obtain. The data is further preprocessed with algorithms of SR Research Data Viewer, which further reduce noise due to e.g. eye tremor and micro-saccades. What we meant by "inherent noise" is (2), the variability in the eye movements of any specific participant over a given text that is not task relevant or cannot be accounted for, and that extracting task relevant signal from eye movements trajectories is challenging. We now clarify this in the revised paper.

---

### Official Review · Reviewer_4c7d · 2025-10-31

**Soundness:** 4
**Presentation:** 4
**Contribution:** 3
**Rating:** 8
**Confidence:** 5

**Summary:**

This paper presents a novel and important investigation into decoding open-ended, text-specific information goals from eye movements during reading. The work is rigorous, introducing a clear task framework (goal selection and reconstruction) and systematically evaluating a range of discriminative and generative models. The experiments are comprehensive, using a large-scale, high-quality dataset (OneStop) and thoughtfully evaluating model generalization. While the generative task remains highly challenging, the strong performance of the best discriminative model, especially on the difficult "same critical span" task, provides compelling evidence that eye movements contain fine-grained information about a reader's goal. This work opens promising new directions for both scientific inquiry and practical applications.

**Strengths:**

1.This is the first work to systematically address the decoding of arbitrary, text-specific information goals from eye movements. It moves beyond previous work that classified pre-defined procedural reading tasks (e.g., reading vs. skim-reading) to a more challenging and practically relevant semantic decoding task. The contribution is significant and opens a new research direction.

2.The experimental design is exemplary. The data splits are carefully constructed to evaluate generalization to "New Participants," "New Texts," and "New Text & Participant," providing a clear and honest assessment of model robustness. The subdivision of the selection task into "Different" and "Same" critical spans offers nuanced insight into task difficulty.

3 The paper provides a thorough benchmark, covering simple heuristics, adapted state-of-the-art discriminative models, and pioneering generative LLM-based approaches. This offers a valuable overview of the landscape for this new task.

4. The evaluation methodology is a strength. Beyond selection accuracy, the authors propose a robust set of metrics for the generative task, including question word/category agreement, BERTScore, and a creative downstream QA accuracy metric, which convincingly demonstrates the utility of the generated questions.

5. The paper is accompanied by a code repository and includes extensive details in the main text and appendices regarding model architectures, hyperparameters, and training procedures, ensuring the work is easily reproducible.

**Weaknesses:**

1. As the results show, the generative task is exceptionally difficult, and model performance, especially on new texts, is still limited. The generated questions are not yet on par with human-composed ones.

2.The paper successfully demonstrates that goals can be decoded, but offers less insight into how or why the models make their decisions from a cognitive perspective. The models remain somewhat black-box.


3. While generalization is tested, the significant performance drop in the "New Text & Participant" regime indicates that model robustness is still limited to the distribution of the training data.

**Questions:**

1.In the RoBERTEye-Fixations model, which achieved the best results, did you perform any ablation studies to understand which specific eye-movement features (e.g., fixation duration, saccade amplitude, regressions) were most critical for its performance, particularly on the challenging "Same Span" task?

2.For the generative DalEye-Llama model, performance drops most significantly in the "New Text" regime. Do you attribute this primarily to the model's difficulty in comprehending the new text content itself, or in associating the novel eye-movement patterns on that text with the question generation process?

3.The paper mentions promising applications in education and assistive technology. Given the current accuracy levels (e.g., ~49% for 3-way selection), what do you see as the minimum performance threshold for such systems to be reliably deployable in real-world scenarios? What are the key next steps to bridge this gap?

---

> ### Author Response · Authors · 2025-12-03
>
> Thank you for your thoughtful and positive review!
> Please find responses to your questions below.
>
>
> > In the RoBERTEye-Fixations model, which achieved the best results, did you perform any ablation studies to understand which specific eye-movement features (e.g., fixation duration, saccade amplitude, regressions) were most critical for its performance, particularly on the challenging "Same Span" task?
>
> Thank you for this suggestion. Following your suggestion, we conducted a feature ablation study for the best performing model, RoBERTEye-Fixations. In this analysis, we ablate each of three eye movement feature groups: fixation level eye movement features (e.g. current fixation duration, next saccade direction), word-level features (reading measures aggregated for each word, e.g. sum of all fixation durations on the word), and linguistic features of the words (e.g. word frequency, surprisal). This allows us to evaluate the contributions of two different eye movement representations (fixation-level and word-level) and the importance of enabling interactions of eye movement features with linguistic word characteristics. We further divide the fixation-level features into two groups: fixations and saccades and evaluate the contribution of each group.   Finally, we also ablate the total paragraph reading time feature, as it is a straightforward measure that doesn’t require an eyetracker. The results are presented below and in Appendix F.3.
>
> | Model                                     | All  | Different Spans | Same Span |
> | -------------------------------------- | :-----: | :--------------: | :---------: |
> | RoBERTEye-Fixations           | 51.2  | 72.8            | 57.5      |
> | - Paragraph Reading Time     | 48.7 | 71.8            | 53.3      |
> | - Text Features                       | 47.0  | 70.5            | 52.3      |
> | - Word Level Features           | 47.7  | 69.8            | 54.4      |
> | - Fixation Level Features       | 49.0  | 70.2            | 56.1      |
> | - Fixation Fixation Features   | 48.5  | 70.6            | 53.7      |
> | - Fixation Saccade Features  | 47.0  | 70.5            | 52.3      |
>
>
> We found that removing any of the feature groups reduces performance across all evaluation settings. Examining the All evaluation, the largest drop is observed when ablating text features, followed by word-level features, then paragraph reading time, and finally fixation-level features. A possible explanation for the relatively large performance decrease when removing the text features is the well-established correspondence between eye movements and the “big three”: word length, predictability, and frequency (Kliegl et al., 2004; Rayner et al., 2004, 2011, among others). Providing models with this information may help them disentangle eye-movement signals that stem from lexical properties (e.g., longer fixations due to longer words) from those that reflect relevance to the information-seeking task.
>
> Surprisingly, removing only the fixation features or the saccade features separately, compared to all the fixation-level features together, degrades performance more in the All and Different Spans evaluations but less in the Same Span evaluation. This suggests that the model relies on different features in each evaluation.
>
> ### References
>
> Kliegl, R., Grabner, E., Rolfs, M., & Engbert, R. Length, frequency, and predictability effects of words on eye movements in reading. European Journal of Cognitive Psychology, 16(1-2), 262–284 (2004).
>
> Rayner, K., Warren, T., Juhasz, B., & Liversedge, S. The Effect of Plausibility on Eye Movements in Reading. Journal of experimental psychology. Learning, memory, and cognition, 30, 1290–301 (2004).
>
> Rayner, K., Slattery, T. J., Drieghe, D., & Liversedge, S. P. Eye movements and word skipping during reading: Effects of word length and predictability. Journal of Experimental Psychology: Human Perception and Performance, 37(2), 514–528 (2011).

---

> ### Author Response · Authors · 2025-12-03
>
> > For the generative DalEye-Llama model, performance drops most significantly in the "New Text" regime. Do you attribute this primarily to the model's difficulty in comprehending the new text content itself, or in associating the novel eye-movement patterns on that text with the question generation process?
>
> Thank you for this question. We believe that this performance drop is rooted in the combination of the New Item evaluation regime difficulty and the question reconstruction task difficulty. In the New Text regime, the model encounters completely new textual materials and questions. This stands in stark contrast to the New Participant regime, where the model has already seen both the text and the possible questions during training, a considerably easier generalization problem. Furthermore, the generation task is inherently more difficult than the discriminative question selection task, where models do generalize well to the New Text regime (see results in Table 2). Thus, we believe that the combination of the task and the evaluation regime lead the Llama model to fail to generalize in this case. We clarify this point in the revised manuscript (Section 6.2).
>
> > The paper mentions promising applications in education and assistive technology. Given the current accuracy levels (e.g., ~49% for 3-way selection), what do you see as the minimum performance threshold for such systems to be reliably deployable in real-world scenarios? What are the key next steps to bridge this gap?
>
> This is a great question. We believe that performance thresholds for deployment in real-world settings should be determined in collaboration with the end-users (e.g. educators), taking into account their specific needs and objectives. Furthermore, this will likely involve moving from in-lab to field data collection settings, and expanding the current research to additional populations such as L2 learners and children.

---

### Official Review · Reviewer_rzyv · 2025-11-03

**Soundness:** 2
**Presentation:** 2
**Contribution:** 2
**Rating:** 4
**Confidence:** 4

**Summary:**

The paper introduces a cognitive-state decoding problem of recovering information-seeking goals (text questions) from a reader’s eye movements while reading a piece of text. The authors propose different task formulations for recovering the stimulus – (1) question selection and (2) question generation. They develop discriminative and generative models for the above tasks, that recover the question given the text and eye movement data. Results shown on the OneStop eye-tracking data showing discriminative models (RoBERTEye-Fixations) perform better than random while generative models (DalEye-Llama) shows promising performance on less challenging cases.

**Strengths:**

The dataset (OneStop) and problem setup is well suited to the objective of recovering reading goals. The evaluation regimes which include splitting data by new participant and new text is well conceived and creation of two tiers of difficulty are useful in comparing model performance in challenging settings.

The authors experiment with different types of baselines – heuristics, discriminative models based on adaptions of prior work and generative LLM models (DalEye-LLaVA, DalEye-Llama).

**Weaknesses:**

1. I would like to see what types of gaze features (eg: fixation durations, word revisits) are more useful for recovering the information seeking goals. Stronger experiments are required to investigate the feature attributions by gradually phasing out these features one by one from the eye movements data to train the models.

2. It is not clear in the paper if the question and the text span containing the corresponding answer have significant substring overlap. If so, the problem becomes more trivial where users just have to look for specific strings from the question in the text. A more realistic and challenging case to evaluate would be if the user would have to understand and infer the meaning from the passage if the language is phrase differently.

3. The authors should more comprehensively discuss about relevant literature in the field of eye movements conditioned on information seeking goals, for instance
- Synthesizing Human Gaze Feedback for Improved NLP Performance (EACL 2023) which generates gaze patterns conditioned on the reader’s intent / task
- GazeXplain: Learning to Predict Natural Language Explanations of Visual Scanpaths (ECCV 2024) which predicts scanpaths during performing visual question answering tasks or when instructed to search for a particular object in the image


Questions / Suggestions for Improvement:
- Show n-gram overlap between target question and corresponding text span and correlate with model correctness.
- More thorough experiments to analyse eye feature importances

**Questions:**

NA

---

> ### Author Response · Authors · 2025-12-03
>
> Thank you very much for your review.
>
> Please find responses to your comments below. We have also made the corresponding revisions in the paper.
>
> > It is not clear in the paper if the question and the text span containing the corresponding answer have significant substring overlap. If so, the problem becomes more trivial where users just have to look for specific strings from the question in the text. A more realistic and challenging case to evaluate would be if the user would have to understand and infer the meaning from the passage if the language is phrase differently.
> > Show n-gram overlap between target question and corresponding text span and correlate with model correctness.
>
> ## Part 1: n-gram overlap
>
> Thank you for providing the opportunity to clarify this point.
>
> The underlying textual materials and questions are from the OneStopQA dataset. These questions were composed such that “the correct answer typically does not appear verbatim in the passage” (Berzak et al. 2020). In other words, in order to answer the question correctly, the reader needs to infer the answer from the text, and thus the task is not extractive QA. We now clarify this aspect of the question answering task in the revised paper (lines 97-99).
>
> Further, following your suggestion, we performed an analysis of the question-text substring overlap in OneStop and compared it to the extractive QA dataset SQuAD. This analysis provides further evidence that the information seeking tasks in OneStop are not string matching tasks. The details and results of this analysis are provided below.
>
>
> ### Substring Overlap Analysis
>
> We computed n-gram overlap between the question and the passage. We measured this overlap using the proportion of matching unigrams (ROUGE-1), bigrams (ROUGE-2), and unigrams restricted to content words, with respect to the number of n-grams in the question (precision), in the text (recall), and the harmonic mean of the two (F1). We further provide a breakdown of the question overlap with the passage into the words within and outside the question's critical portion of the text. We compared this overlap with the corresponding overlap in SQuAD, an extractive QA dataset in which answers appear verbatim in the text. Below, and in the revised paper (Appendix F.1), we present the results of this analysis.
>
> We found that the unigram overlap between OneStop questions and passages (46%) is lower than the corresponding overlap in SQuAD (53%). We similarly find a smaller overlap in OneStop for content words (38% OneStop, 49% SQuAD), and for bigrams (12% OneStop, 20% SQuAD). We note that these comparisons are adequate with the Rouge measures as the OneStop and SQuAD datasets have similar passage lengths (110 OneStop, 120 SQuAD) and question lengths (10 words on average in both). We further found that the n-gram overlap of the questions in OneStop with the critical span text is relatively low (33% for unigrams, 10% bigrams), and only slightly higher than with the text outside of the critical span (28% unigrams, 3% bigrams). This further suggests that the answer text does not differ drastically from non-relevant text in terms of substring overlap with the question. Overall, we find that the overlap in OneStopQA is relatively low, both in absolute terms and compared to SQuAD, providing empirical support for the inferential nature of the information seeking tasks in OneStop.
>
> The n-gram overlap analysis results are presented below. Values in parentheses represent overlap computed based only on content words.
>
> | Dataset     | Text Part     | Rouge-1 P | Rouge-1 R | Rouge-1 F1 | Rouge-2 P | Rouge-2 R | Rouge-2 F1 |
> | ----------- | ------------- | ------------------------- | ------------------------- | -------------------------- | --------- | --------- | ---------- |
> | **SQuAD**  | Paragraph      | 0.527 (0.488)     | 0.071 (0.052)     | 0.122 (0.091)      | 0.208     | 0.021     | 0.037      |
> | **OneStop** | Paragraph       | 0.463  (0.376)   | 0.059 (0.040)    | 0.104 (0.071)      | 0.127     | 0.012     | 0.022      |
> |                     | Critical Span                   | 0.338 (0.285)     | 0.126 (0.097)    | 0.176 (0.137 )    | 0.100     | 0.033     | 0.048      |
> |                     | Out of Critical Span        | 0.283 (0.153)     | 0.050 (0.024)     | 0.084 (0.040)     | 0.034     | 0.005     | 0.008      |

---

> ### Author Response · Authors · 2025-12-03
>
> ## Part 2: Correlation with model correctness
>
> Your comment leads to an intriguing follow-up question: is the task easier when there is more lexical overlap between the question and the text?
>
> To answer this question, we performed a new analysis which examines the correlation between the ngram overlap of the question with the critical span and the probability that the RoBERTEye-Fixations model assigns to the correct question. The analysis is based on the framework from Shubi et al. (2025). It uses a linear mixed-effects model in which the outcome variable is the probability RoBERTEye-Fixations assigns to the correct question, and the main predictor of interest is the lexical overlap between the question and the critical span. The model additionally includes a set of control variables: the reading time before, within, and after the critical span; the paragraph’s position within the experiment; whether the participant answered the comprehension question correctly; the paragraph’s length and difficulty level (original or simplified); and both the starting location and length of the critical span. To account for repeated measurements from the same readers and the same paragraphs, the model also includes random effects for participants and for paragraphs.
>
> Interestingly, we find that the relationship of the question-critical span lexical overlap with model accuracy is not statistically significant, as can be seen in the coefficients table below. We report the complete analysis in Appendix F.2.
>
> | Feature                                   | Coefficient              |
> |:------------------------------------------|:-------------------------:|
> | Reading Time Before Critical Span         | $-0.068^{***}$           |
> | Reading Time In Critical Span             | $0.139^{***}$            |
> | Reading Time After Critical Span          | $-0.073^{***}$           |
> | Paragraph Position                        | $0.011^{**}$             |
> | Answered Correctly                        | $0.010^{*}$              |
> | Paragraph Length                          | $0.029^{***}$            |
> | Paragraph Level (original / simplified)   | $-0.007$   |
> | Critical Span Start Location              | $0.005$    |
> | **Question–Critical Span Lexical Overlap**    | $-0.007$   |

---

> ### Author Response · Authors · 2025-12-03
>
> > I would like to see what types of gaze features (eg: fixation durations, word revisits) are more useful for recovering the information seeking goals. Stronger experiments are required to investigate the feature attributions by gradually phasing out these features one by one from the eye movements data to train the models.
>
> > More thorough experiments to analyse eye feature importances
>
> Thank you for this suggestion. Following your suggestion, we conducted a feature ablation study for the best performing model, RoBERTEye-Fixations. In this analysis, we ablate each of three eye movement feature groups: fixation level eye movement features (e.g. current fixation duration, next saccade direction), word-level features (reading measures aggregated for each word, e.g. sum of all fixation durations on the word), and linguistic features of the words (e.g. word frequency, surprisal). This allows us to evaluate the contributions of two different eye movement representations (fixation-level and word-level) and the importance of enabling interactions of eye movement features with linguistic word characteristics. We further divide the fixation-level features into two groups: fixations and saccades and evaluate the contribution of each group.   Finally, we also ablate the total paragraph reading time feature, as it is a straightforward measure that doesn’t require an eyetracker. The results are presented below and in Appendix F.3.
>
> | Model                                     | All  | Different Spans | Same Span |
> | -------------------------------------- | :-----: | :--------------: | :---------: |
> | RoBERTEye-Fixations           | 51.2  | 72.8            | 57.5      |
> | - Paragraph Reading Time     | 48.7 | 71.8            | 53.3      |
> | - Text Features  | 47.0  | 70.5            | 52.3      |
> | - Word Level Features  | 47.7  | 69.8            | 54.4      |
> | - Fixation Level Features| 49.0  | 70.2            | 56.1      |
> | - Fixation Fixation Features| 48.5  | 70.6            | 53.7      |
> | - Fixation Saccade Features  | 47.0  | 70.5            | 52.3      |
>
>
> We found that removing any of the feature groups reduces performance across all evaluation settings. Examining the All evaluation, the largest drop is observed when ablating text features, followed by word-level features, then paragraph reading time, and finally fixation-level features. A possible explanation for the relatively large performance decrease when removing the text features is the well-established correspondence between eye movements and the “big three”: word length, predictability, and frequency (Kliegl et al., 2004; Rayner et al., 2004, 2011, among others). Providing models with this information may help them disentangle eye-movement signals that stem from lexical properties (e.g., longer fixations due to longer words) from those that reflect relevance to the information-seeking task.
>
> Surprisingly, removing only the fixation features or the saccade features separately, compared to all the fixation-level features together, degrades performance more in the All and Different Spans evaluations but less in the Same Span evaluation. This suggests that the model relies on different features in each evaluation.
>
> > The authors should more comprehensively discuss about relevant literature in the field of eye movements conditioned on information seeking goals
>
> Thank you for these suggestions. We have updated the related work section of the paper to include these papers and two additional references on relevance prediction from eye movements by Bhattacharya et al. (2020a,b). We believe that the current literature review captures relevant work in a comprehensive and adequate manner.

---

> ### Author Response · Authors · 2025-12-03
>
> ## References
>
>
> Berzak, Yevgeni, Jonathan Malmaud, and Roger Levy. "STARC: Structured Annotations for Reading Comprehension." Proceedings of the 58th Annual Meeting of the Association for Computational Linguistics. 2020.
>
> Omer Shubi, Cfir Avraham Hadar, and Yevgeni Berzak. 2025. “Decoding Reading Goals from Eye Movements”. Proceedings of the 63rd Annual Meeting of the Association for Computational Linguistics.
>
> Nilavra Bhattacharya, Somnath Rakshit, and Jacek Gwizdka. 2020. “Towards Real-time Webpage Relevance Prediction Using Convex Hull Based Eye-tracking Features.” ACM Symposium on Eye Tracking Research and Applications.
>
> Nilavra Bhattacharya, Somnath Rakshit, Jacek Gwizdka, and Paul Kogut. 2020. “Relevance Prediction from Eye-movements Using Semi-interpretable Convolutional Neural Networks.” Proceedings of the 2020 Conference on Human Information Interaction and Retrieval.
>
> Kliegl, R., Grabner, E., Rolfs, M., & Engbert, R. Length, frequency, and predictability effects of words on eye movements in reading. European Journal of Cognitive Psychology, 16(1-2), 262–284 (2004).
>
> Rayner, K., Warren, T., Juhasz, B., & Liversedge, S. The Effect of Plausibility on Eye Movements in Reading. Journal of experimental psychology. Learning, memory, and cognition, 30, 1290–301 (2004).
>
> Rayner, K., Slattery, T. J., Drieghe, D., & Liversedge, S. P. Eye movements and word skipping during reading: Effects of word length and predictability. Journal of Experimental Psychology: Human Perception and Performance, 37(2), 514–528 (2011).

---

### Author Response · Authors · 2025-12-03

We thank the reviewers for their positive assessment of our work, and for the constructive suggestions that helped us further strengthen the paper.

The reviewers highlighted the **novelty and significance of the task** (e.g. H8io: “a new research paradigm for the combination of eye movement and language models; The contribution is significant and opens a new research direction”; 4c7d: “first work to systematically address decoding arbitrary, text-specific information goals”), the **strength of our experimental design** (e.g., rzyv: “The dataset (OneStop) and problem setup is well suited to the objective of recovering reading goals. The evaluation regimes which include splitting data by new participant and new text is well conceived”; 4c7d: “The experiments are comprehensive, using a large-scale, high-quality dataset (OneStop) and thoughtfully evaluating model generalization”), and **evidence that eye movements contain meaningful, goal-relevant signal that models can exploit** (e.g., rzyv: “showing discriminative models (RoBERTEye-Fixations) perform better than random while generative models (DalEye-Llama) shows promising performance on less challenging cases.”;H8io: “Results… indicating that eye movements encode semantic goal information”).

We would like to highlight two key analyses that we performed in response to the reviewers' comments. (1) We introduced a mixed-effects **modeling framework that links model performance to cognitively interpretable properties of gaze behavior during goal decoding.** This analysis grounds the work in cognitive theory by connecting goal decoding to psycholinguistic findings on information-seeking reading. (2) We added a **feature ablation study** for the RoBERTEye-Fixations model, quantifying the contribution of different eye-movement features.

Beyond these additions, we have (3) measured the **lexical overlap** between questions and passages (and critical spans), confirming the non-extractive nature of the information seeking task, and further characterized the effect of this overlap on model performance. (4) For the question reconstruction task, we added a comparison with a **fine-tuned GPT-4o-mini** model using the same text+gaze representation as DalEye-Llama. We hope that these new results and the additional clarifications and revisions provided below address the comments raised by the reviewers and further highlight the contributions of our work.

Note to AC: given the exceptional circumstances during the ICLR rebuttal period, we were not able to engage with the reviewers. We would like to stress that all the reviews for the paper are positive, and the comments touch on relatively specific points. Our responses and the revised paper reflect over two weeks of running additional experiments, which we believe comprehensively and thoroughly address the reviewers' comments.

---

### Meta-Review · Area_Chair_UUSi · 2026-01-11

**Summary:**

The three reviewers raised several substantive concerns about the original submission. Reviewer rzyv questioned the contribution of different eye movement features and whether substring overlap between questions and text might trivialize the task. Reviewer H8io highlighted the lack of cognitive interpretation linking gaze behavior to model performance, the unexplained performance drop on unseen texts, and the absence of comparisons with fine-tuned multimodal LLMs. Reviewer 4c7d, while positive overall, also sought ablation studies and cognitive grounding for the results. All reviewers acknowledged the novelty of the task and the strength of the experimental design.

**Reviewer Concerns:**

The authors provided a comprehensive and scientifically rigorous rebuttal that addresses the major concerns raised.

Feature Ablation Study (rzyv, 4c7d): The authors conducted a thorough ablation of the RoBERTEye-Fixations model, examining fixation-level features, word-level features, text features, and paragraph reading time. This analysis reveals meaningful patterns, such as the importance of text features for disentangling lexical effects from goal-relevant signals, directly addressing both reviewers' requests.

Lexical Overlap Analysis (rzyv): The n-gram overlap analysis comparing OneStop to SQuAD demonstrates that OneStop has lower question-passage overlap (46% vs. 53% unigrams), confirming the inferential rather than extractive nature of the task. The additional mixed-effects analysis showing no significant correlation between overlap and model accuracy further strengthens this point.

Cognitive Interpretation (H8io, 4c7d): The newly added mixed-effects modeling framework connects model performance to psycholinguistically interpretable features. The finding that longer reading times within critical spans improve performance while reading times outside them hurt performance is a substantive contribution that grounds the work in cognitive theory.

Fine-tuned LLM Comparison (H8io): The authors fine-tuned GPT-4o-mini using the same input representation, showing comparable results to DalEye-Llama. This addresses the concern about baseline comparisons with state-of-the-art models.

Performance Drop Explanation (H8io, 4c7d): The authors provide a reasonable explanation attributing the drop to the combination of task difficulty (generation) and evaluation regime difficulty (completely new materials), noting they found no evidence of overfitting in learning curves.

Outstanding concerns

The inherent difficulty of the generative task on unseen texts remains a limitation, though this reflects the genuine challenge of the problem rather than a methodological flaw. The explanation for the performance drop, while reasonable, remains somewhat speculative without further investigation.

**Reviewer Scores:**

Reviewer rzyv (4): All three stated concerns were directly addressed with substantial new experiments. The comprehensive ablation study, lexical overlap analysis with comparison to SQuAD, and expanded literature review should satisfy this reviewer. I anticipate a score increase to 5.

Reviewer 4c7d (8): This reviewer was already positive and their questions were addressed satisfactorily. The ablation study and cognitive grounding further strengthen the paper. Score likely maintained at 8.

Reviewer H8io (4): The cognitive interpretation analysis directly addresses their primary concern. The fine-tuned GPT-4o-mini comparison and explanation for the performance drop address the remaining concerns. I expect a score increase to.

Rationale for Acceptance

This paper makes a significant contribution by introducing a novel task at the intersection of eye-tracking research and language modeling. The experimental design is careful, with thoughtful generalization evaluations across participants and texts. The rebuttal demonstrates scientific rigor, with over two weeks of additional experiments addressing reviewer concerns comprehensively. With one solidly positive reviewer and the remaining concerns substantively addressed, the paper meets the bar for acceptance.

---

### Decision · Program_Chairs · 2026-01-26

Accept (Poster)